# Maturation of persistent and hyperpolarization-activated inward currents shapes the differential activation of motoneuron subtypes during postnatal development

Simon A Sharples, Gareth B Miles*

School of Psychology and Neuroscience, University of St Andrews, St Andrews, United Kingdom

**Abstract** The size principle underlies the orderly recruitment of motor units; however, motoneuron size is a poor predictor of recruitment amongst functionally defined motoneuron subtypes. Whilst intrinsic properties are key regulators of motoneuron recruitment, the underlying currents involved are not well defined. Whole-cell patch-clamp electrophysiology was deployed to study intrinsic properties, and the underlying currents, that contribute to the differential activation of delayed and immediate firing motoneuron subtypes. Motoneurons were studied during the first three postnatal weeks in mice to identify key properties that contribute to rheobase and may be important to establish orderly recruitment. We find that delayed and immediate firing motoneurons are functionally homogeneous during the first postnatal week and are activated based on size, irrespective of subtype. The rheobase of motoneuron subtypes becomes staggered during the second postnatal week, which coincides with the differential maturation of passive and active properties, particularly persistent inward currents. Rheobase of delayed firing motoneurons increases further in the third postnatal week due to the development of a prominent resting hyperpolarization-activated inward current. Our results suggest that motoneuron recruitment is multifactorial, with recruitment order established during postnatal development through the differential maturation of passive properties and sequential integration of persistent and hyperpolarization-activated inward currents.

*For correspondence:
gbm4@st-andrews.ac.uk

**Competing interest:** The authors declare that no competing interests exist.

## Editor's evaluation

This manuscript will be of interest to those studying the neuroscience of movement as it addresses a fundamental aspect of movement: motoneuron recruitment. The authors use spinal cord slices to provide a comprehensive analysis of motoneuron intrinsic properties, passive and active, that mature in the early postnatal period in mice and coincide with their differentiation into 'slow' and 'fast' phenotypes. They argue that these properties together shape motoneuron recruitment, suggesting that textbook views of orderly recruitment may be oversimplifications.

## Introduction

Fine control of muscle force is a prerequisite for the generation of complex movement. This fine control relies on the orderly recruitment of 'motor units'; the smallest functional unit of the motor system, consisting of alpha motoneurons and the muscle fibres they innervate (*Liddell and Sherrington, 1924*). Motor units can be subdivided based on the twitch kinetics of their muscle fibres.

Different motor units are recruited in order of their force-generating capacity and susceptibility to fatigue, with slow fatigue resistant, fast fatigue resistant, and fast fatigable motor units recruited in sequence (*Burke et al., 1973*; *Henneman, 1957*). This orderly recruitment of motor units has been accounted for by the size principle (*Burke et al., 1973*; *Henneman, 1957*; *Somjen et al., 1965*), with smaller motor units recruited prior to larger motor units. Although motoneurons that make up different subtypes of motor units exist on a size-based continuum (*Burke et al., 1982*), there is considerable overlap in geometrical and physiological measures of size between motoneuron subtypes, such that motoneuron size appears to be a poor predictor of recruitment order (*Gustafsson and Pinter, 1984*; *Zengel et al., 1985*). It has therefore been argued that recruitment order is more heavily influenced by the functional subtype of a given motor unit and not strictly dependent on physical size (*Burke et al., 1973*; *Cope and Clark, 1991*; *Cope and Pinter, 1995*; *Zajac and Faden, 1985*). Here we address whether motoneuron recruitment is purely dependent on the size principle or whether functional classes of spinal motoneurons possess distinct electrophysiological properties that help ensure orderly recruitment of motoneuron subtypes.

The intrinsic excitability of neurons is influenced by their compartmental, geometric structures, which are endowed with specialized complements of ion channels that support flexible output (*Brocard, 2019*; *Deardorff et al., 2021*; *Heckman et al., 2008*). For example, nonlinearities in the input-output relationship of motoneurons are produced by persistent inward currents (PICs) generated by persistent sodium and L-type calcium channels located on the axon initial segment (AIS) and distal dendrites, respectively (*Brocard et al., 2016*; *Chatelier et al., 2010*; *Hounsgaard et al., 1988*; *Hounsgaard and Kiehn, 1993*; *Lee and Heckman, 1998a*; *Li and Bennett, 2003*; *Quinlan et al., 2011*; *Schwindt and Crill, 1980*). Variations in the complement of ion channels expressed by subtypes of spinal motoneurons have been demonstrated and their roles in producing nonlinear firing behaviours described (*Bos et al., 2018*; *Brocard, 2019*; *Heckman et al., 2007*; *Soulard et al., 2020*). Many of these currents are activated below the spike threshold, meaning their differential expression is also likely to contribute to orderly recruitment across motoneuron subtypes (*Zhang and Dai, 2020*).

To address the role of specific ion channels in motoneuron subtype recruitment, we studied differences in the intrinsic properties and underlying currents that contribute to the activation of putative fast and slow motoneurons, which were identified based on delayed and immediate firing profiles during whole-cell patch-clamp electrophysiological recordings in lumbar spinal cord slices of postnatal mice (*Bhumbra and Beato, 2018*; *Bos et al., 2018*; *Durand et al., 2015*; *Leroy et al., 2015*). Activation currents, otherwise known as rheobase currents, were measured as a proxy for motoneuron recruitment. We used postnatal development as a model to dissect the relative importance of the intrinsic properties of motoneurons to the differential activation of motoneuron subtypes, which contributes to the orderly recruitment of motoneurons required for the gradation of muscle force and fine motor control. Postnatal development in the rodent provides an opportune system to address this issue given that animals are largely sessile during the first postnatal week, produce rudimentary weight-bearing, hindlimb-based locomotion during week 2 (*Brocard et al., 1999*), and complex locomotor behaviours consistent with adult animals by week 3 (*Altman and Sudarshan, 1975*). It would therefore be expected that key properties that establish orderly recruitment of functionally defined motoneuron subtypes would manifest in parallel with the emergence of complex motor behaviour during postnatal development.

We identify key properties and underlying currents of motoneuron subtypes and pinpoint when these properties contribute to the functional differentiation of motoneuron subtypes, particularly with respect to their rheobase currents, during postnatal development. We find that during the first postnatal week lumbar motoneurons are activated according to passive properties predictive of size; however, delayed and immediate firing subtypes are functionally homogeneous, leading to overlap in rheobase currents across motoneuron subtypes. Rheobase current of delayed and immediate firing motoneurons then becomes staggered by the second postnatal week, which coincides with an increase in the size of delayed firing motoneurons and a more depolarized activation of sodium PICs in delayed firing motoneurons. Rheobase current becomes staggered further during week 3 as delayed firing motoneurons develop a prominent hyperpolarization-activated inward current (Ih) that is active at resting membrane potential and further increases rheobase, possibly by providing a depolarizing shunt. Taken together, our data demonstrate that motoneuron recruitment is multifactorial and support the notion that fast motoneurons are not simply scaled-up versions of slow motoneurons

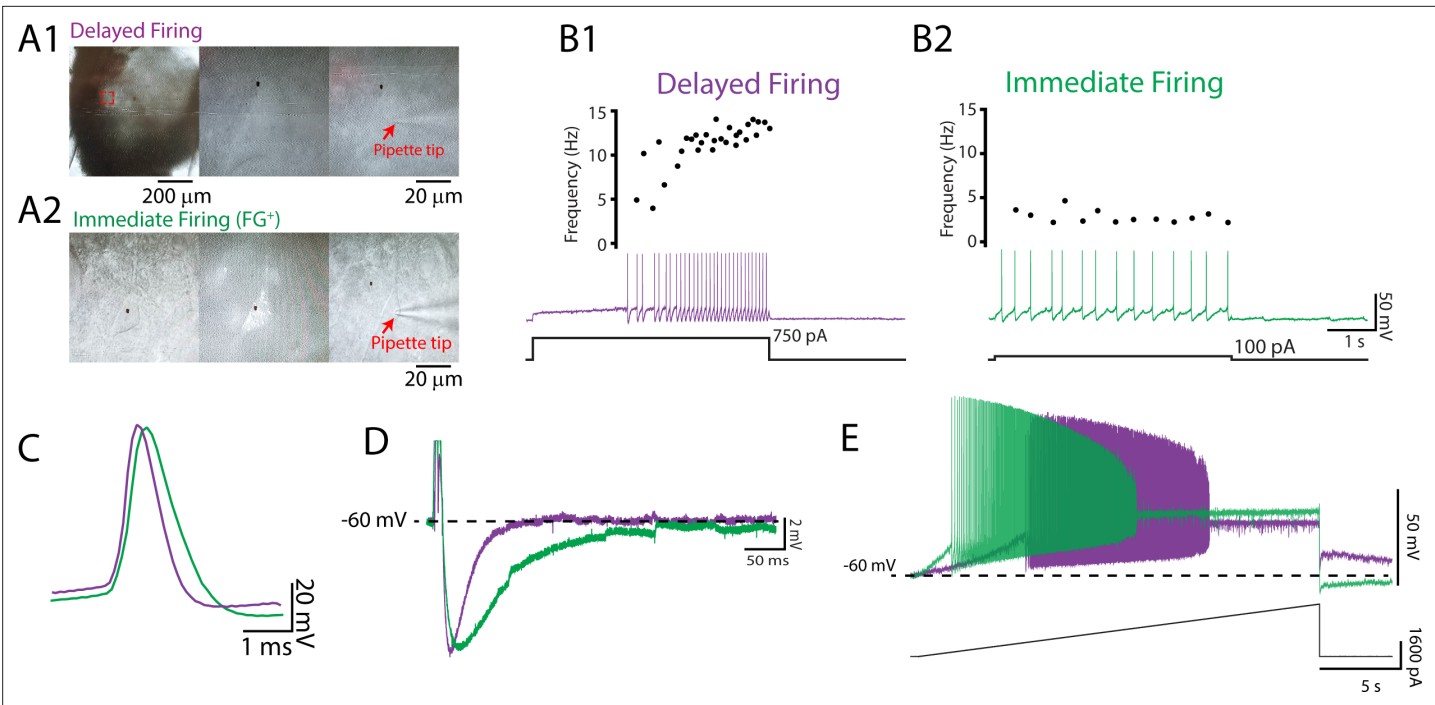

**Figure 1.** Delayed and immediate firing motoneurons exhibit intrinsic properties consistent with fast- and slow-type motoneurons, respectively. (**A**) Whole-cell patch-clamp recordings were obtained from motoneurons identified in the ventrolateral horn of the lumbar spinal cord with a subset retrogradely labelled with Fluoro-Gold injected intraperitoneally. Two motoneuron subtypes were identified based on the onset latency for repetitive firing and subsequent changes in firing rates during a 5 s depolarizing current step applied at rheobase (**B**): a delayed onset of repetitive firing and an accelerating firing rate (**B1**; purple) and an immediate onset for repetitive firing with a stable or adapting firing rate (**B2**; green). Delayed and immediate firing motoneuron subtypes display intrinsic properties that are consistent with those of fast- and slow-type motoneurons illustrated by broader single-action potentials (**C**), longer medium afterhyperpolarizations (**D**), and lower rheobase currents (**E**) in immediate compared to delayed firing motoneurons.

(**Kernell and Zwaagstra, 1981**). Whilst recruitment is influenced by size, activation properties of inward currents, which are differentially expressed in motoneuron subtypes, ensure orderly recruitment of functionally defined subtypes of spinal motoneurons.

## Results
### Motoneuron subtypes diversify functionally during postnatal development

We used whole-cell patch-clamp electrophysiology to define the intrinsic properties that differentiate motoneuron subtypes and support orderly recruitment. We adopted an established electrophysiological approach to identify two motoneuron subtypes in vitro based on their repetitive firing profiles in response to the injection of a 5 s depolarizing square current pulse at rheobase (**Bhumbra and Beato, 2018**; **Bos et al., 2018**; **Durand et al., 2015**; **Leroy et al., 2015**). The motoneuron subtypes that were identified included a delayed repetitive firing profile with an accelerating firing rate (**Figure 1A1, B1**) and an immediate repetitive firing profile with a steady or adapting firing rate (**Figure 1A2 and B2**). We identified delayed and immediate firing motoneurons at all ages studied (P1–20) with intrinsic properties that are consistent with those of fast and slow motoneurons, respectively (**Figure 1C–E**; **Supplementary file 1**; **Gardiner, 1993**; **Gustafsson and Pinter, 1984**; **Leroy et al., 2014**; **Martínez-Silva et al., 2018**). Our measurements of intrinsic properties are in line with previous reports from mouse lumbar motoneurons at similar postnatal stages (**Leroy et al., 2014**; **Nakanishi and Whelan, 2010**; **Quinlan et al., 2011**; **Smith and Brownstone, 2020**). Postnatal development in mice is highly stereotyped, with hindlimb weight-bearing emerging during the second postnatal week and complex locomotor behaviours emerging after the start of the third (**Altman and Sudarshan, 1975**). We therefore grouped motoneurons by week to explore broad changes in intrinsic properties of motoneuron

subtypes between behaviourally defined milestones across motor development. Principal component analysis (PCA) was first deployed to identify global changes in intrinsic properties of motoneuron subtypes during development given the high dimensionality (25 variables) and large sample size (N = 261 motoneurons, 95 animals) of our data (*Figure 2*). This allowed us to focus subsequent analyses on variables that covary across motoneuron type during postnatal development.

PCA identified six principal components (PCs) that accounted for greater than 75% of variance; 48% of which was accounted for by PC1 and PC2 (*Figure 2A*). PC1 accounted for 35.5% of variance and, as would be expected, had the greatest loading scores from rheobase current (0.82) and passive properties such as input resistance (–0.83) and membrane time constant (–0.7), but a relatively low weight from whole-cell capacitance (0.54), which is related to cell surface area (*Gustafsson and Pinter, 1984*; *Taylor, 2012*; *Figure 2B*). Analysis of PC scores for PC1 revealed differential maturation in delayed and immediate firing motoneurons whereby PC1 scores increased in delayed but did not change in immediate firing motoneurons, with differences between delayed and immediate firing motoneurons revealed at weeks 2 and 3, but not week 1 (*Figure 2D*; $F_{(2,187)}$ = 14.1, p=2.1e-6). PC2 accounted for 14.0% of variance, had the greatest loading from action potential threshold voltage (0.77), which can also influence rheobase (*Figure 2B*), and matured in parallel in delayed and immediate firing motoneurons. PC2 scores were larger in slow motoneurons in week 1, decreasing in both subtypes, but to a greater extent in immediate firing motoneurons, such that PC2 scores were more negative in immediate compared to delayed firing motoneurons in week 3 (*Figure 2E*; $F_{(2,228)}$ = 8.6, p=2.5e-4). PC3 accounted for 9.6% of variance and had the greatest loading score derived from repetitive firing properties such as maximum firing rate (0.78) and frequency-current gain within the sub-primary range (0.73) (*Figure 2C*). PC3 scores decreased in immediate firing motoneurons only between weeks 2 and 3 (*Figure 2F*; $F_{(2,228)}$ = 21.9, p=1.7e-9). Afterhyperpolarization (AHP) half width is a strong predictor of motoneuron type (*Gustafsson and Pinter, 1984*), and in line with this, scored high for PC1 (–0.6) but relatively lower for PC2 (0.5) and PC3 (0.3). Further analysis revealed an increase in clustering of PCs 1–3 in delayed and immediate firing motoneurons as a function of development (*Figure 2G–I*), indicating that these two populations functionally diverge during postnatal development. PCs 4–6 accounted for less variance within our data (PC4: 7.4%; PC5: 5.5%; PC6: 4.8%) than PCs 1–3 and did not significantly change during development or reveal differences between motoneuron subtypes.

## Postnatal diversification of intrinsic properties shapes the rheobase of motoneuron subtypes

PCA indicated that rheobase current is a key predictor of the two motoneuron subtypes and can account for the greatest variance across postnatal development. Rheobase, defined as the minimal current to elicit repetitive firing of action potentials, was measured during slow (100 pA/s) depolarizing current ramps in addition to long (5 s) depolarizing current steps that were used to identify motoneuron types. Both measures of rheobase were strongly correlated at all ages studied (*Figure 3—figure supplement 1A–C*; W1: r = 0.87, p<1.0e-15; W2: r = 0.95, p<1.0e-15; W3: r = 0.92, p<1.0e-15) and demonstrated similar changes during postnatal development (*Figure 3—figure supplement 1D and E*). We therefore next set out to determine intrinsic properties that mature in parallel, contribute to changes in rheobase current measured with depolarizing current ramps during postnatal development, and may therefore contribute to the establishment of orderly recruitment of motoneuron subtypes.

As would be expected based on the size principle, passive properties contributed substantially to the loading scores within PC1. Rheobase current and passive properties did not differ between delayed and immediate firing motoneurons during the first postnatal week (*Figure 3D–G*); however, rheobase current was correlated with passive properties such as capacitance and input resistance, irrespective of motoneuron subtype, at this point in time (*Figure 3H1–I1*). This finding is supportive of an early activation scheme that is based on size rather than functionally defined subtypes of motoneurons.

The rheobase current then increased in delayed but not immediate firing motoneurons between weeks 1 and 2 (*Figure 3D*; $F_{(2,247)}$ = 12.1, p=9.6e-6). The nonuniform maturation of rheobase currents between motoneuron subtypes led to a 57% increase in the rheobase current range (W1: 909 pA; W2: 1596 pA) and a decrease in the recruitment gain across the samples of motoneurons studied (*Figure 3E*). In line with what would be predicted based on the size principle, we found a parallel increase in whole-cell capacitance (*Figure 3F*; $F_{(2,253)}$ = 3.6, p=0.03) and a reduction in input resistance

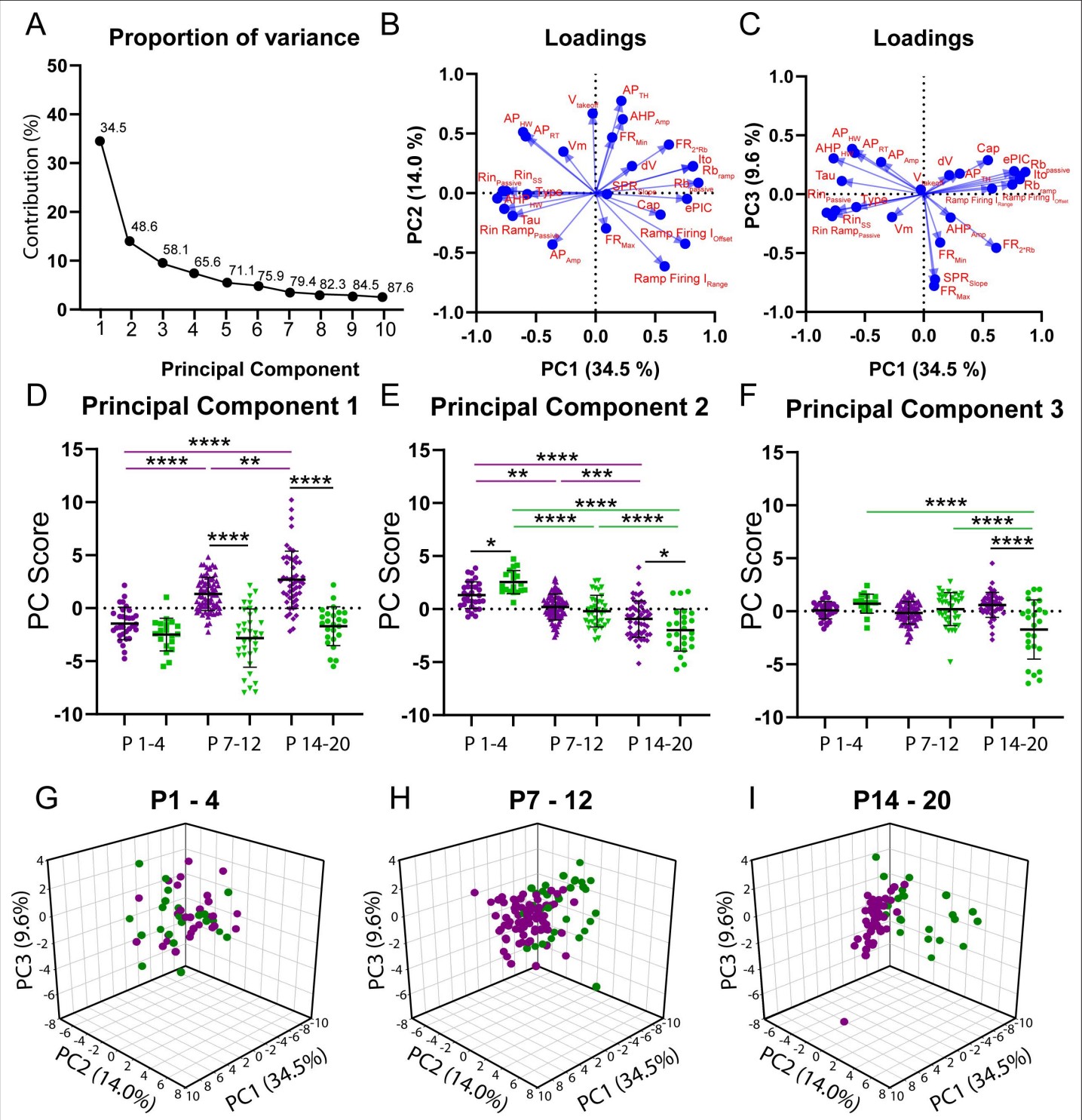

**Figure 2.** Principal component analysis (PCA) reveals divergent maturation of the intrinsic properties of delayed and immediate firing motoneuron subtypes during postnatal development. (**A**) Scree plot illustrating 10 principal components (PCs) identified in the PCA, with the first six accounting for greater than 75% of variance. (**B**) Variable loadings for PC1 and PC2 with arrow length representing loading score for each variable denoted in red text. (**C**) Variable loadings for PC1 and PC3 with arrow length representing loading score for each variable (red). (**D–F**) PC scores for PCs1–3 for delayed (purple) and immediate firing (green) motoneurons across the first three weeks of postnatal development. Individual PC scores are displayed for each cell, black bars represent mean ± SD. Statistical analysis was conducted using a two-way ANOVA and Holm–Sidak post hoc analysis. Asterisks denote significant differences from pairwise comparisons *p<0.05, **p<0.01, ***p<0.001, ****p<0.0001. (**G - I**) 3D scatterplots between PCs 1, 2, and 3 for delayed (purple) and immediate firing (green) motoneurons across weeks 1–3.

*Figure 2 continued on next page*

*Figure 2 continued*

The online version of this article includes the following figure supplement(s) for figure 2:

**Source data 1.** Eigenvalues (*Figure 2A*), loading scores (*Figure 2B-C*), principal component scores (*Figure 2D-F*), and scatterplot data (*Figure 2G-I*) derived from principal component analysis on 25 intrinsic properties studied from motoneuron subtypes across postnatal development.

(*Figure 3G*; $F_{(2,253)}$ = 10.6, p=3.9e-5) in delayed but not immediate firing motoneurons between weeks 1 and 2. As a result, rheobase currents of delayed and immediate firing motoneurons become functionally staggered during week 2, whereby delayed firing motoneurons have a significantly higher rheobase current, whole-cell capacitance, and lower input resistance than immediate firing motoneurons. The membrane time constant increased between weeks 1 and 2 in immediate but not delayed firing motoneurons, leading to a longer time constant in immediate compared to delayed firing motoneurons during week 2 (*Supplementary file 1*; $F_{(2,253)}$ = 7.9, p=4.7e-4). Together these results indicate that maturation of motoneuron passive properties contribute to the functional specification of motoneuron subtypes during the second postnatal week.

The rheobase current increased further in delayed but not immediate firing motoneurons between weeks 2 and 3 (*Figure 3D*; $F_{(2,247)}$ = 12.1, p=9.6e-6), leading to a 182% increase in rheobase current range compared to week 2 and a 320% increase compared to week 1 (W3: 2906 pA) and a further decrease in the recruitment gain (*Figure 3E*). However, the increase in rheobase current in week 3 was not paralleled by further changes in passive properties. In fact, the relationship between rheobase current and capacitance appears to become weaker over this time period (*Figure 3H*: W1: $r^2$ = 0.35; W2: $r^2$ = 0.27; W3: $r^2$ = 0.25). This result suggests that rheobase current, and its maturation between weeks 2 and 3, may not be solely dependent on passive properties, as would be predicted by the size principle.

## PICs contribute to the rheobase of delayed and immediate firing motoneurons but not their maturation between weeks 2 and 3

Changes in the membrane potential of motoneurons in response to slow depolarizing current ramps are characterized by two phases: an initial linear phase, presumably predominated by passive properties, and an accelerating depolarization of the membrane potential (*Figure 4A*), mediated by the activation of PICs (*Delestrée et al., 2014*; *Iglesias et al., 2011*; *Kuo et al., 2006*). We find evidence for both phases in delayed and immediate firing motoneurons at all ages (P1–20). PICs have been extensively studied in spinal motoneurons and underlie nonlinear firing behaviours (*Heckman et al., 2007*; *Schwindt and Crill, 1980*), but their relative contribution to rheobase has been less explored. We therefore next tested the possibility that PICs may have a larger impact on the trajectory of the membrane potential in immediate firing motoneurons compared to that of delayed firing motoneurons and may therefore shape their rheobase at stages when our data suggest that differential recruitment cannot be governed by size-based principles alone.

The influence of PICs on rheobase was first estimated by calculating the difference between the measured rheobase current and an estimate of what the rheobase current would be if the membrane depolarization followed a linear trajectory determined by passive properties alone (*Figure 4A*, $I_{passive}$ – $I_{actual}$) (*Delestrée et al., 2014*; *Jørgensen et al., 2021*). This estimate of PIC is denoted ePIC and did not differ between delayed and immediate firing motoneurons in week 1, increased in delayed firing motoneurons only in weeks 2 and 3, and was larger in delayed compared to immediate firing motoneurons at both time points (*Figure 4B*: $F_{(2,242)}$ = 10.2, p=5.7e-5). We also assessed the influence of PICs on motoneuron output during triangular depolarizing current ramps as an additional and well-established estimate of PIC magnitude (*Harvey et al., 2006a*; *Hounsgaard et al., 1988*; *Quinlan et al., 2011*; *Schwindt and Crill, 1977*; *Figure 4—figure supplement 1*). In this assay, a lower current at derecruitment on the descending limb of the triangular ramp compared to the recruitment current on the ascending limb (negative Delta I) is evidence of a PIC (*Harvey et al., 2006c*; *Hounsgaard and Kiehn, 1985*). Delta I did not differ between delayed and immediate firing motoneurons during week 1, became more negative (increased PIC) in delayed firing motoneurons during week 2, but did not change further into week 3 (*Figure 4—figure supplement 1*; $F_{(2,165)}$ = 6.1, p=0.003).

The underlying PICs were next measured directly in voltage clamp during slow (10 mV/s) depolarizing voltage ramps (*Figure 4E and F*; *Lee and Heckman, 1998a*; *Quinlan et al., 2011*) to account

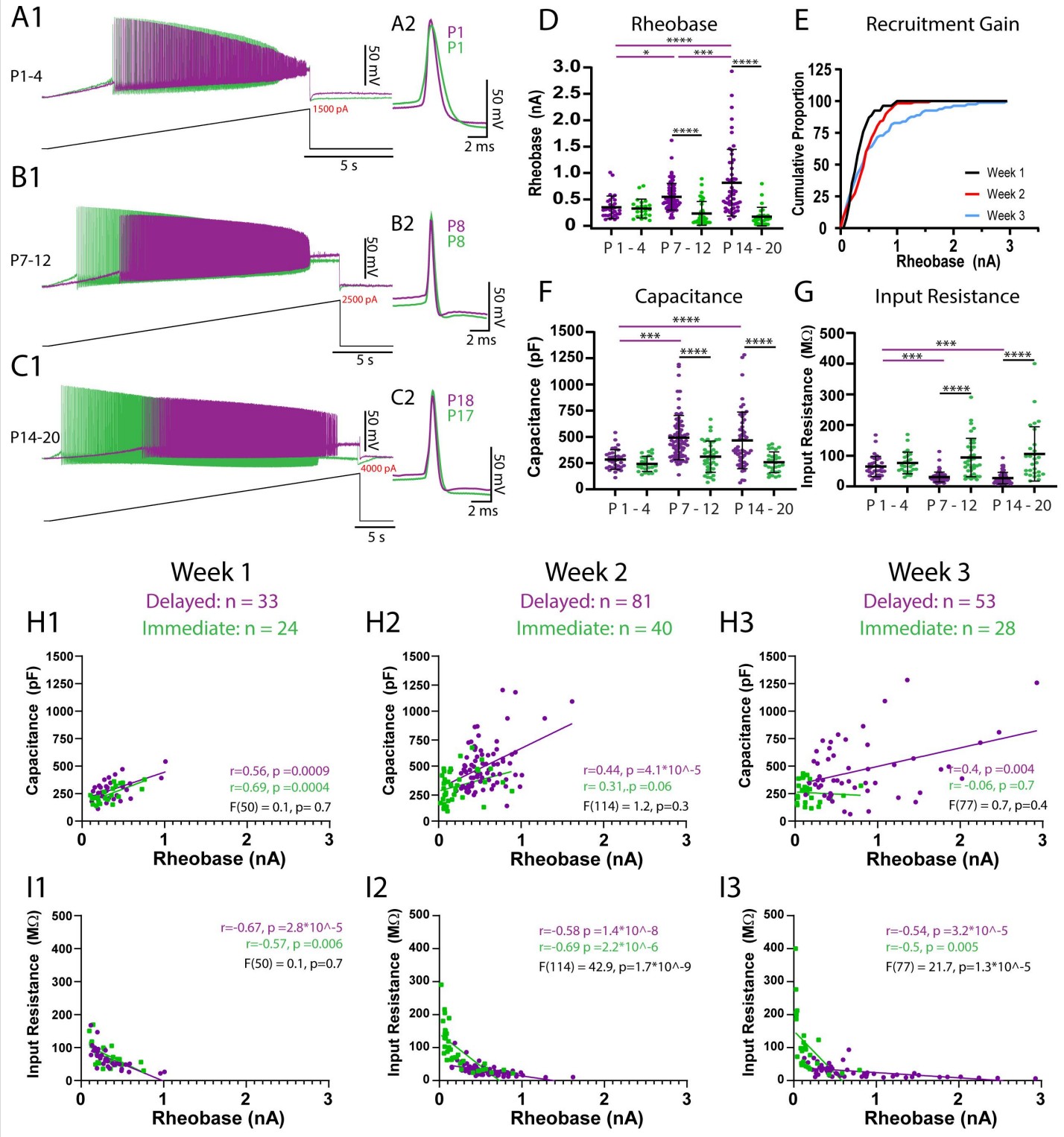

**Figure 3.** Passive properties contribute to maturation of rheobase currents between weeks 1 and 2 but additional factors contribute from weeks 2–3. The rheobase current of delayed (purple) and immediate firing (green) motoneurons was assessed using slow (100 pA/s) depolarizing current ramps. (**A1–C1**) Representative traces of rheobase and repetitive firing of delayed and immediate firing motoneurons at weeks 1–3. (**A2–C2**) First action potential produced upon recruitment for delayed and immediate firing motoneurons at weeks 1–3. (**D**) Rheobase increases in delayed but not immediate firing motoneurons across weeks 1–3. (**E**) Cumulative proportion histograms of rheobase currents of motoneurons sampled during weeks 1 (black), 2 (red), and 3 (blue) indicate a progressive decrease in the recruitment gain. (**F**) Whole-cell capacitance increases, and input resistance decreases

*Figure 3 continued on next page*

*Figure 3 continued*

(**G**) in delayed but not immediate firing motoneurons between weeks 1 and 2 but not further into week 3. Individual data points are displayed, black bars represent mean ± SD. Statistical analysis was conducted using a two-way ANOVA and Holm–Sidak post hoc analysis. Asterisks denote significant differences from pairwise comparisons *p<0.05, **p< 0.01, ***p<0.001, ****p<0.0001. Scatterplots of rheobase current and passive properties (**H1-H3**: whole-cell capacitance; **I1-I3**: input resistance) (*Gustafsson and Pinter, 1984*) for delayed (purple) and immediate firing (green) motoneurons during postnatal weeks 1–3. Pearson correlations were performed for each type at each time point with outcomes displayed in purple text for delayed and green text for immediate firing motoneurons. A simple linear regression comparing the regression slopes for each motoneuron subtype at each time point was performed to determine the degree of similarities between subtypes, with outcomes displayed in black text.

The online version of this article includes the following figure supplement(s) for figure 3:

**Source data 1.** Rheobase (*Figure 3D*), recruitment gain (*Figure 3E*), capacitance (*Figure 3F*), and input resistance (*Figure 3G*) values, and correlations (*Figure 3H-I*) from motoneuron subtypes studied across postnatal development.

**Figure supplement 1.** Rheobase is correlated and shows similar changes during postnatal development when measured during depolarizing current ramps or square depolarizing current steps.

**Figure supplement 1—source data 1.** Rheobase measured from motoneuron subtypes across development on depolarizing ramps (**A**) and current steps (**B**) are correlated during the first (**C**), second (**D**), and third (**E**) postnatal weeks.

---

for the possibility that current clamp-based estimates of PIC may not be purely dependent on PICs. In line with our two estimates of PICs measured in current clamp, the amplitude of the PIC measured in voltage clamp was significantly larger in delayed compared to immediate firing motoneurons at weeks 2 and 3 (*Figure 4G*; $F_{(1,61)}$ = 17.9, p=7.8e-5), which can be accounted for by the larger size (higher capacitance) of delayed compared to immediate firing motoneurons given that PIC density (current/ whole-cell capacitance) did not differ between subtypes (*Supplementary file 2*). PIC amplitude did not change in either subtype between weeks 2 and 3 (*Figure 4G*; $F_{(1,61)}$ = 0.02, p=0.9) and was not correlated with rheobase current at either age (W2: $r$ = –0.1, p=0.5; W3: $r$ = –0.2, p=0.2).

Differences in the magnitude of PICs between delayed and immediate firing motoneurons do not appear to explain differences in rheobase of motoneuron subtypes or changes that we observe during development; in fact, they might support a lower rheobase in delayed firing motoneurons. We therefore considered other factors that may contribute to differences in rheobase, such as the voltage at which PICs first activate. We found no difference in the voltage at which depolarization of the membrane potential began to accelerate, in response to slow depolarizing current ramps, between motoneuron subtypes during week 1. However, the onset of the acceleration phase became hyperpolarized in both motoneuron subtypes in week 2 and continued to hyperpolarize even further in the immediate firing motoneurons into week 3, becoming significantly more hyperpolarized compared to delayed firing motoneurons (*Figure 4C*; $F_{(2,247)}$ = 6.1, p=0.003). Interestingly, acceleration onset voltage and spike threshold matured in parallel (*Figure 4D*), such that the amplitude of the depolarizing acceleration from onset to spike threshold was similar between delayed and immediate firing motoneurons and maintained at all ages (*Supplementary file 2*; $F_{(2,247)}$ = 2.2, p=0.1). In line with estimates of PICs in current clamp, PIC onset voltage measured in voltage clamp was more depolarized in delayed compared to immediate firing motoneurons at both weeks 2 and 3 ($F_{(1,61)}$ = 57.6, p=2.2e-10), did not change between weeks 2 and 3 in either motoneuron type (*Figure 4H*; $F_{(1,61)}$ = 0.2, p=0.7), but was correlated with rheobase current at both ages (W2: $r$ = 0.4, p=0.007; W3: $r$ = 0.7, p=1.3e-5). Interestingly, PIC activation in immediate firing motoneurons occurs relatively close to their respective resting membrane potential (RMP: W2: –58.3 ± 5.4; W3: –61.5 ± 6.7 mV) which may facilitate their recruitment (*Supplementary file 1* and *Supplementary file 2*).

PICs are produced by a combination of ion channels that conduct sodium and calcium currents, with sodium PICs activated at more hyperpolarized voltages than calcium PICs (*Li and Bennett, 2003*; *Quinlan et al., 2011*). We deployed pharmacological tools to block Nav1.6 channels (*Brocard et al., 2016*; *Krzemien et al., 2000*; *Li and Bennett, 2003*; *Schaller and Caldwell, 2000*) and L-type calcium channels (*Anelli et al., 2007*; *Bouhadfane et al., 2013*; *Hounsgaard and Kiehn, 1993*; *Li et al., 2004*; *Perrier and Hounsgaard, 1999*; *Zhang et al., 2006*), which conduct a part of the sodium and calcium PICs, respectively, to determine the relative contribution of each type of PIC to rheobase of motoneuron subtypes. Blocking Nav1.6 channels with the selective blocker, 4,9-anhydrotetrodotoxin (4,9-AH-TTX: 200 nM) produced a 31% reduction in PIC amplitude in delayed firing motoneurons and a 60% reduction in PIC amplitude in immediate firing motoneurons (*Figure 4J*; delayed: n = 22 MNs, 11 animals; immediate: n = 12 MNs, 7 animals; $F_{(1,29)}$ = 22.9, p=4.7e-5). Blocking L-type calcium channels

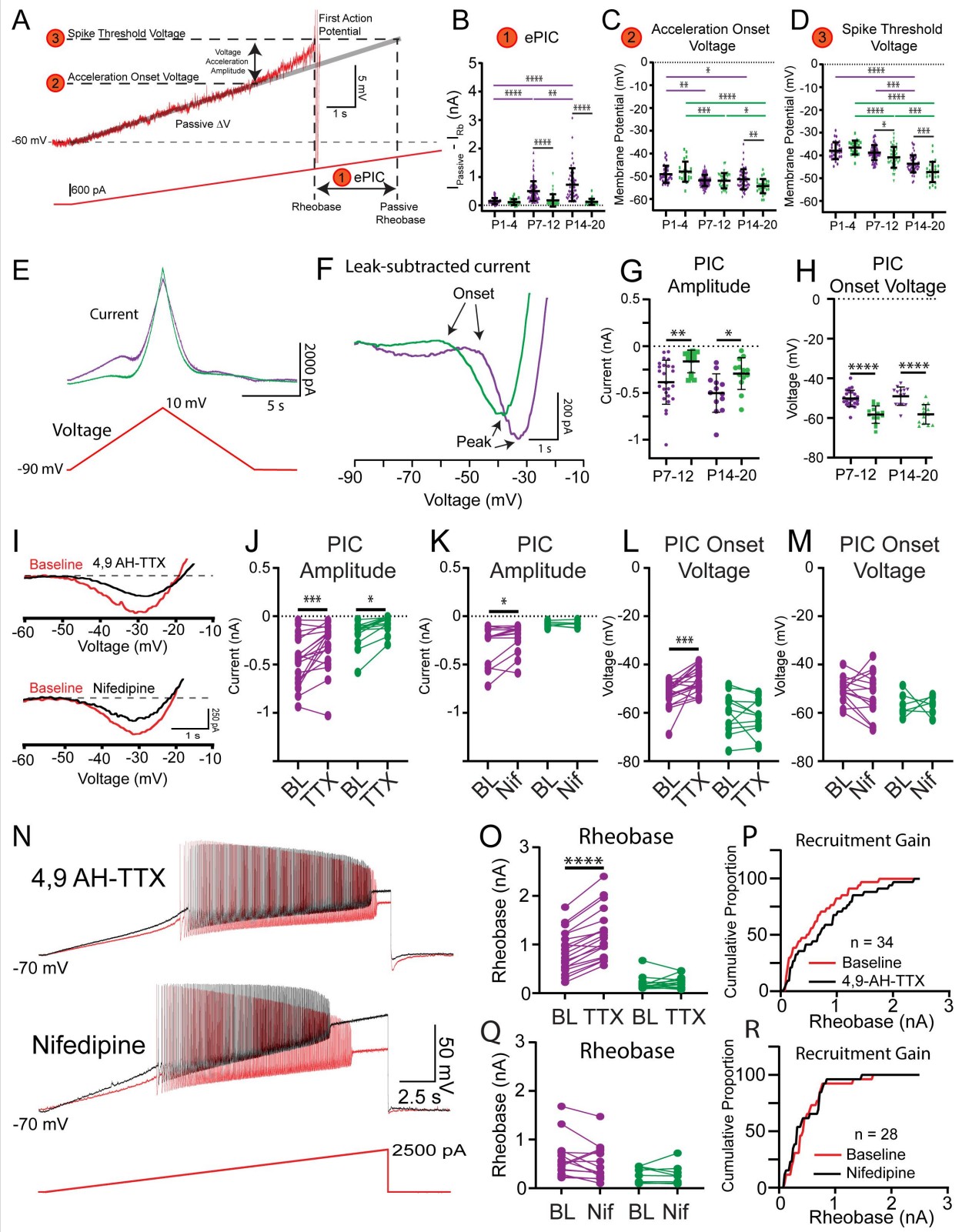

**Figure 4.** Nav1.6-conducting persistent inward currents (PICs) shape motoneuron activation. (**A**) Rheobase was assessed using slow depolarizing current ramps (100 pA/s from –60 mV). During these ramps, the trajectory of the membrane potential approaching the spike threshold is characterized by a passive, linear phase (DV: grey line) and an accelerating phase. The impact of the underlying current producing the membrane potential acceleration, termed ePIC (**A1, B**), is estimated as the difference between the extrapolated passive and actual rheobase currents. The acceleration onset voltage (**A2,**

*Figure 4 continued on next page*

*Figure 4 continued*

C) is the voltage at which the membrane potential deviates by greater than 1 mV from the linear function fitted to the initial (passive) depolarization of the membrane potential during the ramp. The amplitude of this acceleration phase is calculated from the onset voltage to the threshold of the first action potential (**A3, D**). (**E**) PICs were measured in voltage clamp from delayed (purple) and immediate (green) firing motoneurons of 2-week-old (n = 25 delayed; n = 14 immediate) and 3-week-old (n = 14 delayed; n = 13 immediate) mice using a slow depolarizing voltage ramp (10 mV/s; –90 to –10 mV). (**F**) PIC onset and amplitude were measured from leak-subtracted, low-pass filtered (5 Hz Bessel) traces. (**G**) PICs were larger in amplitude in delayed compared to immediate firing motoneurons at both 2 and 3 weeks. (**H**) PIC onset was significantly more depolarized in delayed compared to immediate firing motoneurons and did not change between weeks 2 and 3. (**I**) Representative traces of a PIC measured in voltage clamp from a delayed firing motoneuron. The relative contribution of Nav1.6 and L-type calcium channels to PICs in delayed and immediate firing motoneurons was tested with 4,9-anhydrotetrodotoxin (4,9 AH-TTX; 200 nM, delayed: n = 22; immediate: n = 12) and nifedipine (20 μM, delayed: n = 16; immediate: n = 10), respectively (black traces). 4,9 AH-TTX reduced PIC amplitude of delayed and immediate firing motoneurons (**J**), whereas nifedipine only reduced the PIC amplitude of delayed firing motoneurons (**K**). 4,9 AH-TTX depolarized PIC onset voltage of delayed firing motoneurons (**L**); nifedipine did not change PIC onset voltage in either subtype (**M**). (**N**) Representative traces from depolarizing current ramps of delayed firing motoneurons used to investigate changes in rheobase before (red) and after (black) blocking Nav1.6 or L-type calcium channels. (**O**) Blocking Nav1.6 increased the rheobase of delayed but not immediate firing motoneurons. (**P**) Cumulative proportion histograms for rheobase currents of delayed and immediate firing motoneurons sampled before (red) and after (black) blocking Nav1.6 channels. Blocking L-type calcium channels did not alter rheobase in either motoneuron subtype (**Q**) or change the recruitment gain (**R**). Individual data points are displayed, black bars represent mean ± SD. Statistical analysis was conducted using a two-way ANOVA and Holm–Sidak post hoc analysis. Asterisks denote significant differences from pairwise comparisons *p<0.05, **p<0.01, ***p<0.001, ****p<0.0001.

The online version of this article includes the following figure supplement(s) for figure 4:

**Source data 1.** Estimated persistent inward current (PIC) (*Figure 4B*), acceleration onset voltage (*Figure 4C*), and spike threshold (*Figure 4D*) measured during depolarizing current ramps.

**Figure supplement 1.** Maturation of recruitment-derecruitment and firing hysteresis during postnatal development.

**Figure supplement 1—source data 1.** Recruitment-derecruitment (**C–E**) and firing rate hysteresis (**G**) measured from motoneuron subtypes across postnatal development.

with nifedipine (20 μM) produced a 22% reduction in PIC amplitude in delayed firing motoneurons but did not change PIC amplitude in immediate firing motoneurons (*Figure 4K*; delayed: n = 16 MNs, 9 animals; immediate: n = 10 MNs, 7 animals; $F_{(1,18)}$ = 6.4, p=0.02). Blocking Nav1.6 channels depolarized the PIC onset voltage (*Figure 4L*; $F_{(1,28)}$ = 15.7, p=4.6e-4), and spike threshold (BL: –34.8 ± 6.0 mV; 4,9-AH-TTX: –26.9 ± 6.3 mV; $F_{(1,29)}$ = 14.2, p=6.6e-4, Holm–Sidak post hoc p=5.9e-11) of delayed firing motoneurons, which translated to an increase in their rheobase current (*Figure 4N and O*; $F_{(1,29)}$ = 33.9, p=2.6e-6). 4,9-AH-TTX also depolarized the spike threshold of immediate firing motoneurons (BL: –45.2 ± 5.7 mV; 4,9-AH-TTX: –42.1 ± 8.5 mV; Holm–Sidak post hoc p=0.01); however, it did not alter their PIC onset voltage (*Figure 4L*; Holm–Sidak post hoc p=0.14) or rheobase current (*Figure 4O*; Holm–Sidak post hoc p=0.47). The differential control of rheobase between motoneuron subtypes led to an overall decrease in the recruitment gain across the population of motoneurons studied (*Figure 4P*). Blocking L-type calcium channels with nifedipine did not change PIC onset voltage (*Figure 4M*; $F_{(1,18)}$ = 0.31, p=0.58), rheobase (*Figure 4N and Q*; $F_{(1,20)}$ = 1.2, p=0.3), or spike threshold (Del: BL: –38.2 ± 7.5 mV; Nif: –38.4 ± 10.1 mV; Imm: BL: –45.2 ± 7.4 mV; Nif: –42.2 ± 6.6 mV; $F_{(1,25)}$ = 1.0, p=0.3) in delayed or immediate firing motoneurons, and as a result, recruitment gain was unaltered (*Figure 4R*).

In these experiments, we set out to determine roles for active properties in shaping motoneuron rheobase and its maturation during postnatal development. These data indicate that PIC activation threshold is more important for shaping rheobase than the overall PIC amplitude, with different complements of ion channels contributing to the PICs of different motoneuron subtypes. Further, rheobase current is not purely dependent on input resistance and is also influenced by properties of currents that are activated in the subthreshold voltage range. Thus, in addition to the maturation of passive properties , differences in the activation properties of PICs may contribute to the staggering or rheobase currents amongst motoneuron subtypes in week 2. However, it is unlikely that PICs contribute to further increases in the rheobase current of delayed firing motoneurons into week 3, given that PIC onset voltage, which is the most important property of the PIC for rheobase, does not change between weeks 2 and 3.

## Hyperpolarization-activated inward currents shape the rheobase of delayed and immediate firing motoneurons after week 3

We next considered other active properties that might contribute to the increase in rheobase current in delayed firing motoneurons between weeks 2 and 3 and further stagger rheobase currents of motoneuron subtypes. HCN channels produce a hyperpolarization-activated inward current (Ih) and are key for producing rebound bursting following inhibition (*Dougherty and Kiehn, 2010*; *Engbers et al., 2011*; *Wilson et al., 2005*; *Zhong et al., 2010*). In some neurons, HCN channels are active near resting potential, producing an inward current that counteracts the hyperpolarizing influence of potassium leak currents to maintain the resting membrane potential (*Bayliss et al., 1994*; *Kiehn et al., 2000*; *Kjaerulff and Kiehn, 2001*; *McLarnon, 1995*; *Perrier et al., 2003*; *Picton et al., 2018*). Putative Ih-mediated sag potentials have been described in larger spinal motoneurons of adult cats and rats (*Araki et al., 1961*; *Burke and ten Bruggencate, 1971*; *Gustafsson and Pinter, 1984*; *Ito and Oshima, 1965*; *Jørgensen et al., 2021*; *Manuel et al., 2009*; *Meehan et al., 2010*; *Takahashi, 1990*; *Zengel et al., 1985*), and have been suggested to decrease the medium AHP (mAHP); a key distinguishing feature of motoneuron subtypes. We therefore hypothesized that an increase in Ih in delayed but not immediate firing motoneurons during week 3 may become important for shaping delayed firing motoneuron rheobase and therefore further refining orderly recruitment.

In line with our hypothesis, we found a prominent increase in depolarizing sag potentials measured in current clamp (*Figure 5A–D*) and underlying Ih measured in voltage clamp (*Figure 5E–G*) in delayed firing motoneurons at week 3, which was abolished by the selective HCN channel blocker, ZD7288 (10 µM: *Figure 5H–J*). The overall sag conductance was estimated (gSag) by calculating the inverse of the I-dV slope (conductance: g) at initial and steady-state potentials during hyperpolarizing current steps and then compared between motoneuron subtypes across development. gSag did not differ between motoneuron subtypes during weeks 1 and 2 but increased in delayed firing motoneurons at week 3 such that it was larger in delayed compared to immediate firing motoneurons at this point in time (*Figure 5B*; $F_{(2,249)}$ = 5.2, p=0.006). Depolarizing sag potentials (*Figure 5C*; $F_{(6,103)}$ = 28.4, p<1.0e-15) and underlying Ih amplitude (*Figure 5F and G*; $F_{(1,61)}$ = 6.1, p=0.01) were larger in delayed compared to immediate firing motoneurons during week 3. Notably, we find a prominent Ih in delayed firing motoneurons at –70 mV (*Figure 5F*; $F_{(1,61)}$ = 13.4, p=0.005), which is near the resting potential of delayed firing motoneurons (Ih –70 mV W3: –119 ± 103 pA; RMP W3: –66.8 ± 3.6 mV), and could also be blocked by ZD7288 (*Figure 5I*; $F_{(1,17)}$ = 8.7, p=0.009). The ZD7288-sensitive current was significantly larger in delayed compared to immediate firing motoneurons at week 3 when measured at both –70 mV (Del: –134 ± 101 pA, Imm: –17 ± 16 pA; $t_{(17)}$ = 3.2, p=0.005) and –110 mV (Del: –1111 ± 803 pA, Imm: –452 ± 246 pA; $t_{(16)}$ = 2.2, p=0.04). Differences in Ih at week 3 are not due to differences in soma size as Ih density was larger in delayed compared to immediate firing motoneurons at week 3 (*Supplementary file 2*; $F_{(4,142)}$ = 6.5, p=7.7e-5), but not week 2 (*Supplementary file 2*; $F_{(4,100)}$ = 0.3, p=0.9). These data suggest that Ih increases and becomes activated at more depolarized voltages in delayed firing motoneurons by the third postnatal week.

Given that we find evidence for Ih becoming activated at more depolarized voltages in delayed firing motoneurons during week 3, we sought to determine if Ih might be active at resting membrane potential. We find a prominent h-current in voltage clamp mode during voltage steps from a holding potential of –50 mV to the respective resting potential of each delayed firing motoneuron (n = 13; Ih Amp = –96 ± 60 pA; Ih dens = –0.17 ± 0.08 pA/pF), which was not observed during voltage steps to resting potential in immediate firing motoneurons (n = 11; Ih Amp = –0.13 ± 11 pA, Ih dens = –0.007 ± 0.06 pA/pF). In addition, ZD7288-sensitive sag potentials were found in delayed but not immediate firing motoneurons during hyperpolarizing current steps that brought the membrane potential from –60 mV to their respective resting membrane potentials (*Figure 6A and B*; $F_{(1,19)}$ = 23.7, p=1.1e-4). Interestingly, blocking Ih with ZD7288 hyperpolarized the resting membrane potential of both delayed and immediate firing motoneurons (*Figure 6C and D*: $F_{(1,20)}$ = 11.0, p=0.003); however, the magnitude of this change was greater in delayed firing motoneurons (*Figure 6E*: $t_{(20)}$ = 3.3, p=0.003). Together these findings indicate that Ih is indeed active at resting potential in delayed but not immediate firing motoneurons.

Following the establishment of Ih at and around resting potential, we next set out to determine if Ih could influence the activation of delayed firing motoneurons. Ih measured at –70 mV was larger in motoneurons with a higher rheobase current (r = - 0.86, p=6.4e-7) and ZD7288 decreased the rheobase

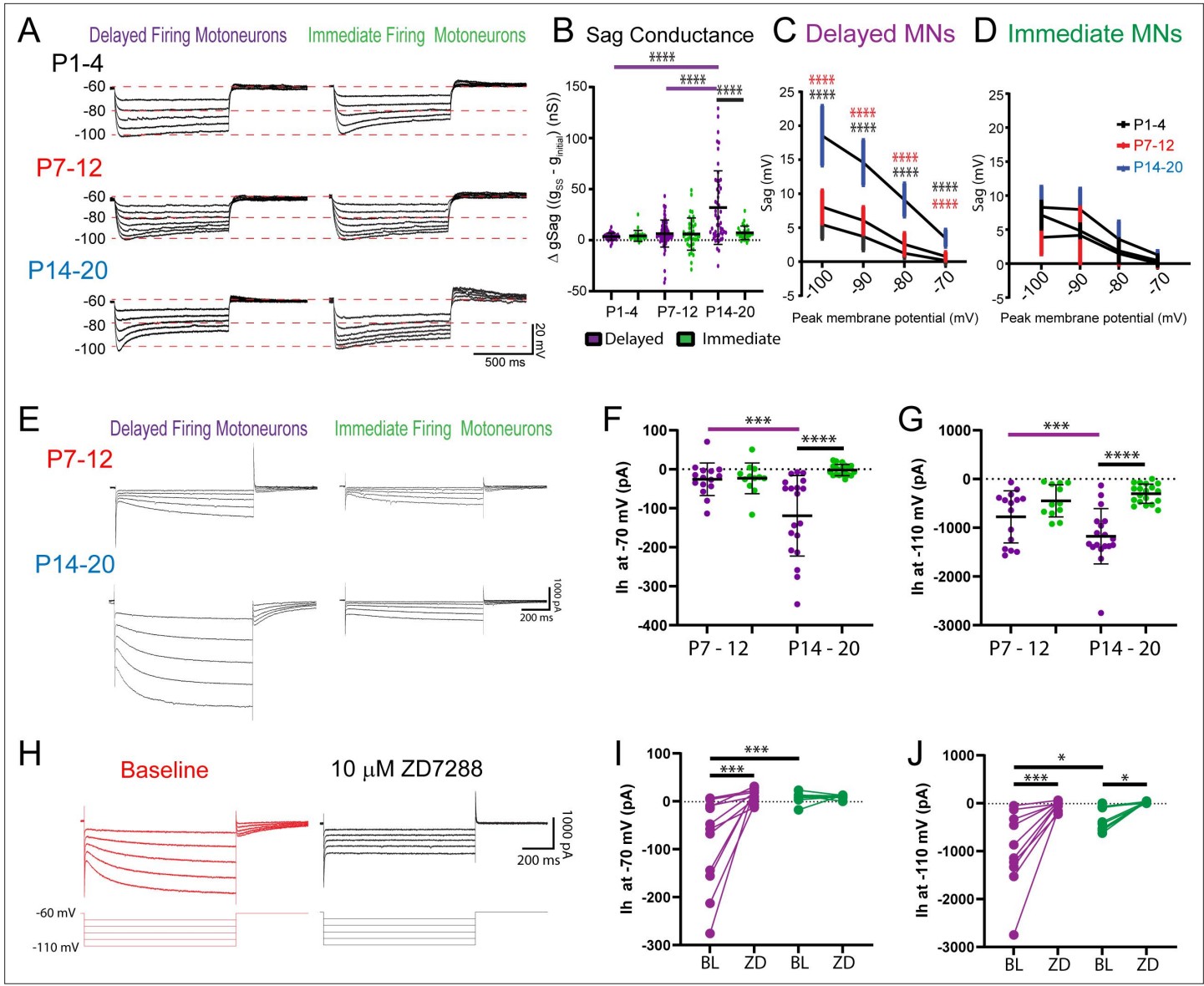

**Figure 5.** Ih increases and becomes activated at more depolarized potentials in delayed firing motoneurons by week 3 . (**A**) Sag potentials were measured from delayed (purple) and immediate firing (green) motoneurons of mice aged P1–4 (n = 54), P7–12 (n = 122), and P14–20 (n = 81). (**B**) The sag conductance (gSag) was estimated (eSag Conductance) during a series of hyperpolarizing current steps (1 s duration). Mean ± SD sag potentials plotted as a function of trough of membrane potential for delayed (**C**) and immediate firing (**D**) motoneurons at P1–4 (black), P7–12 (red), and P14–20 (blue). (**E**) Ih was measured in delayed and immediate firing motoneurons during weeks 2 (Delayed: n = 15; Immediate: n = 12) and 3 (Delayed: n = 19; Immediate: n = 19) in voltage clamp using a series of incremental hyperpolarizing voltage steps (–60 to –110 mV, 1 s duration, 10 mV increments). Ih measured at –70 mV (**F**) and –110 mV (**G**) increased in delayed firing motoneurons between weeks 2 and 3 and was larger in delayed compared to immediate firing motoneurons at week 3 (**H**). Representative trace of Ih recorded in voltage clamp from a delayed firing motoneuron before (red trace) and after (black trace) bath application of the HCN channel blocker ZD7288 (10 µM). ZD7288 blocked Ih in delayed firing motoneurons measured at –70 mV (**I**) and Ih measured at –110 mV in both delayed (n = 13) and immediate firing (n = 9) motoneurons (**J**). Data are presented as individual points for each cell studied. Statistical analyses were conducted using two-way ANOVA and Holm–Sidak post hoc analysis when significant effects were detected. Asterisks denote significant differences from Holm–Sidak post hoc analysis *p<0.05, **p<0.01, ***p<0.001, ****p<0.0001.

The online version of this article includes the following figure supplement(s) for figure 5:

**Source data 1.** Depolarizing sag potentials measured in current clamp from motoneuron subtypes (*Figure 5B-D*) and underlying Ih measured in voltage clamp (*Figure 5F-G*) from motoneuron subtypes at postnatal weeks 2 and 3.

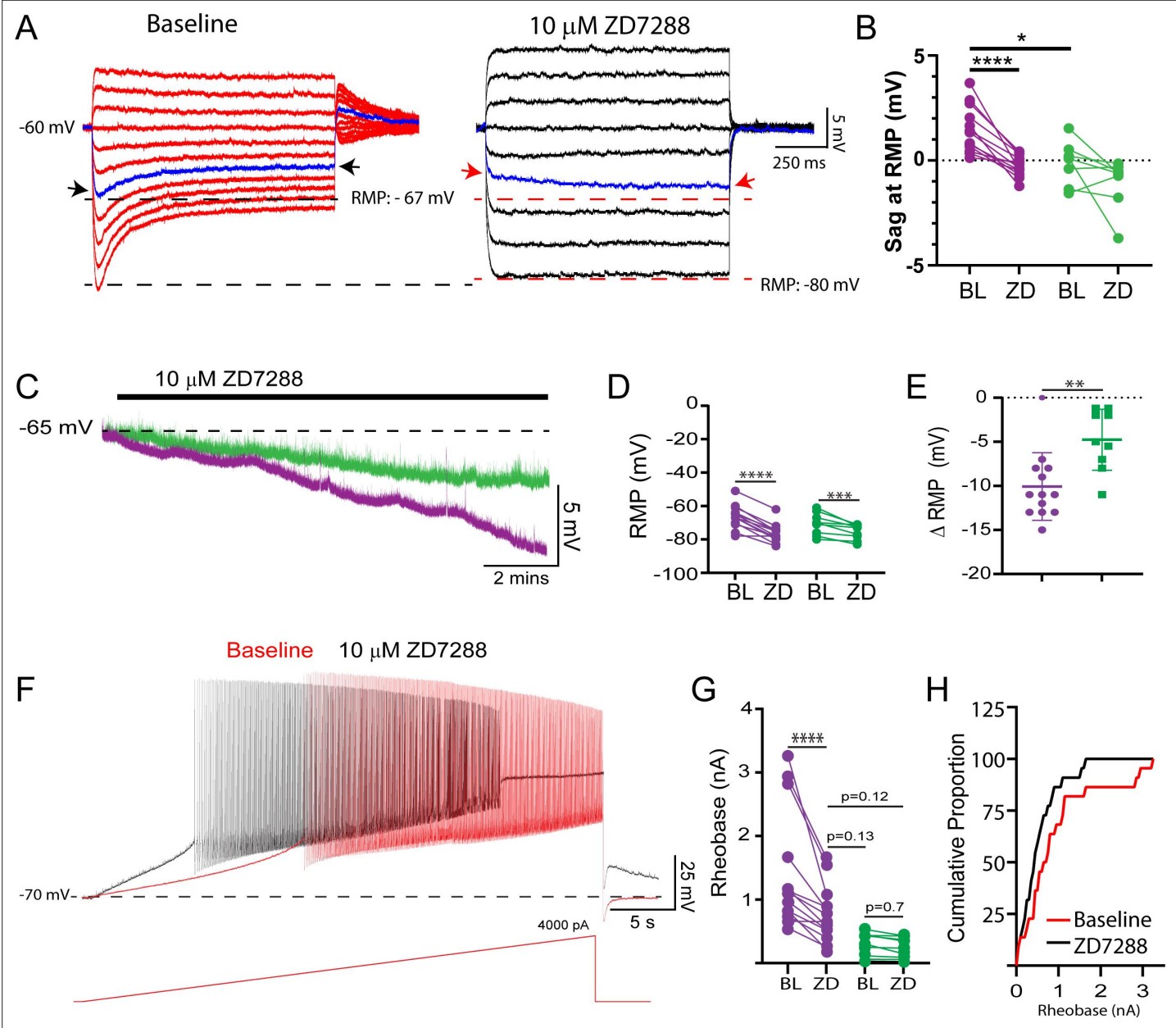

**Figure 6.** A resting Ih increases rheobase of delayed firing motoneurons at week 3. (**A**) Representative trace of depolarizing sag potentials in a delayed firing motoneuron during negative current steps that brought the membrane potential from –60 mV to resting potential (–67 mV) for this cell, which could be blocked by the HCN channel blocker ZD7288 (black trace). The individual blue traces and arrows highlight a depolarizing sag from resting potential at baseline which turns into a slow hyperpolarizing potential following application of ZD7288. (**B**) ZD7288-sensitive depolarizing sag potentials were found at resting membrane potential (RMP) in delayed (n = 13; purple) but not immediate (n = 9; green) firing motoneurons. (**C**) Representative traces of the RMP of delayed and immediate firing motoneurons from lumbar slices of P14–20 mice during blockade of HCN channels with ZD7288. (**D**) ZD7288 hyperpolarized the RMP of both delayed and immediate firing motoneurons; however, the magnitude of the hyperpolarization was larger in delayed firing motoneurons (**E**: unpaired t-test). (**F**) Representative voltage trace of a delayed firing motoneuron during a slow depolarizing (100 pA/s) current ramp before (red) and after (black) ZD7288. (**G**) ZD7288 reduced the rheobase current of delayed but not immediate firing motoneurons. Data are presented as individual points for each cell studied. Statistical analyses were conducted using two-way ANOVA and Holm–Sidak post hoc analysis when significant effects were detected. Asterisks denote significant differences from Holm–Sidak post hoc analysis *p<0.05, **p<0.01, ***p<0.001, ****p<0.0001. (**H**) Recruitment gain represented by cumulative proportion histograms of rheobase currents of motoneurons sampled before (red) and after (black) blocking HCN channels.

The online version of this article includes the following figure supplement(s) for figure 6:

**Source data 1.** Depolarizing sag measured at resting potential (*Figure 6B*), resting membrane potential (*Figure 6D-E*), rheobase (*Figure 6G*), and recruitment gain (*Figure 6H*) before and after ZD7288.

current of delayed but not immediate firing motoneurons (*Figure 6F and G*: $F_{(1,20)}$ = 5.4, p=0.0002), to a level where rheobase current of delayed firing motoneurons was not significantly different from that of immediate firing motoneurons at their respective baseline (*Figure 6G*: Holm–Sidak post hoc: p=0.19). This nonuniform modulation of rheobase currents across motoneuron subtypes led to a 50% reduction in the rheobase current range across the sample of motoneurons studied (BL: 3,227 pA; ZD7288: 1624 pA) and an increase in the recruitment gain (*Figure 6H*). ZD7288 did not change the voltage threshold of the first action potential in either motoneuron subtype (Del: BL: –38.8 ± 3.9, ZD: –37.8 ± 7.4 mV; Imm: –45.0 ± 4.1, ZD: –45.2 ± 3.4 mV; $F_{(1,20)}$ = 0.6, p=0.5).

The effects of ZD7288 on motoneuron intrinsic properties were reproduced with a second HCN channel blocker, ivabradine (10 µM; n = 8 Del; n = 3 Imm), which produced a 45 ± 18% reduction in Ih measured at –110 mV in delayed and immediate firing motoneurons ($F_{(1,9)}$ = 11.5, p=0.008). This reduction in Ih by ivabradine led to a hyperpolarization of the resting membrane potential of both delayed (ΔRMP = –8.0 ± 4.4 mV) and immediate (ΔRMP = –4.3 ± 1.2 mV) firing motoneurons ($F_{(1,8)}$ = 21.2, p=0.002). Ivabradine produced a 26 ± 6% reduction in rheobase of delayed firing motoneurons ($F_{(1,8)}$ = 11.7, p=0.009, Holm–Sidak post hoc p=1.8e-4) but did not significantly change the rheobase of immediate firing motoneurons (Holm–Sidak post hoc p=0.6) or spike threshold in either subtype ($F_{(1,8)}$ = 1.0, p=0.3).

Together these results indicate that delayed firing motoneurons express a prominent Ih that is active at resting membrane potential and can shape their activation. Further, differential maturation of Ih in delayed versus immediate firing motoneurons may contribute to the expansion of the activation range from postnatal weeks 2–3.

## Discussion

We set out to address whether motoneuron recruitment, which we assayed based on the current required to elicit repetitive firing (rheobase), is purely dependent on passive properties, as would be predicted by the size principle, or whether functional classes of spinal motoneurons possess distinct electrophysiological properties that help ensure orderly recruitment of motoneuron subtypes. We used postnatal development of the mouse as a model to dissect the relative importance of motoneuron intrinsic properties to the establishment of orderly recruitment during the emergence of fine motor control. The rate at which motoneurons within a motor pool are recruited is important because it defines the recruitment gain and influences the relative degree of precision or vigour of movement (*Cope and Pinter, 1995*; *Kernell and Hultborn, 1990*; *Nielsen et al., 2019*). We reveal novel roles for specific ion channels in shaping motoneuron subtype recruitment and demonstrate sequential integration of these intrinsic properties during postnatal development. These findings are of broad importance given that orderly recruitment of motoneurons is conserved across phyla (*Ampatzis et al., 2013*; *Azevedo et al., 2020*; *Bawa et al., 1984*; *Burke et al., 1973*; *Gabriel et al., 2011*; *Gardiner, 1993*; *Gustafsson and Pinter, 1984*; *Henneman, 1957*; *Leroy et al., 2015*; *Leroy et al., 2014*; *Manuel et al., 2009*; *Martínez-Silva et al., 2018*; *McLean et al., 2007*; *Zwart et al., 2016*), and differences in the intrinsic properties of motoneuron subtypes lead to increased susceptibility to degeneration in disease (*Fuchs et al., 2013*; *Kaplan et al., 2014*; *Leroy et al., 2014*; *Martínez-Silva et al., 2018*; *Nijssen et al., 2017*) subsequently producing motor dysfunction as a consequence of alterations in recruitment gain within motor pools (*Dengler et al., 1990*; *Hu et al., 2015*; *Thomas et al., 2014*).

We identified putative fast- and slow-type motoneurons based on delayed and immediate repetitive firing profiles respectively. This approach has been previously validated using the molecular markers chondrolectin and matrix metalloproteinase-9 for fast motoneurons, and estrogen-related receptor beta (ERRβ) for slow-type motoneurons (*Enjin et al., 2010*; *Kaplan et al., 2014*; *Leroy et al., 2014*). In line with the initial studies using this approach (*Leroy et al., 2014*), we find that at similar ages (P7–14) delayed firing motoneurons have a lower input resistance, higher rheobase, more depolarized spike threshold, shorter action potentials, and shorter mAHPs compared to immediate firing motoneurons. Given the consistencies between the electrophysiological properties of motoneuron subtypes in our study and those previously reported and validated by others (*Leroy et al., 2014*; *Martínez-Silva et al., 2018*), it is likely that the two subtypes that we studied represent fast and slow motoneurons. However, we cannot rule out the possibility that gamma motoneurons were included in our samples given the lack of clear physiological markers to identify this subtype. A table comparing motoneuron intrinsic properties across studies can be found in *Supplementary file 3*.

Our results suggest that motoneuron recruitment is multifaceted, with multiple layers that establish orderly recruitment of functionally defined spinal motoneuron subtypes. The size principle forms a foundation with recruitment directed by physical properties of neurons. We find that the rheobase current is driven primarily by the physical properties of neurons during the first postnatal week, as would be predicted by the size principle, but rheobase is not linked to the functional subtypes of motoneurons. A steep recruitment gain within a motor pool, with no differentiation between the recruitment of motoneuron subtypes, may be important to facilitate the rapid, ballistic movements, which are typically observed in newborn mice prior to the onset of finer motor control. These findings are consistent with those in human babies where rapid movements are supported by synchronized recruitment of motor unit subtypes to accommodate for the slower and more homogenous muscle twitch properties at early postnatal stages (*Del Vecchio et al., 2020*), which have also been reported in neonatal rodents (*Close, 1964*). Synchronized motoneuron recruitment may also support ongoing maturation of spinal circuits and neuromuscular junctions during early postnatal development (*Hanson and Landmesser, 2004*) when poly-neuronal innervation of muscle fibres (*Bennett et al., 1983*; *Brown et al., 1976*), in addition to gap junction (*Chang et al., 1999*; *Personius et al., 2007*; *Walton and Navarrete, 1991*) and glutamatergic coupling (*Bhumbra and Beato, 2018*) between motoneurons, remains prevalent.

Rheobase currents of motoneuron subtypes become staggered during the second postnatal week, when finer motor control develops, such as hindlimb weight-bearing during stepping. This separation of rheobase coincided with the diversification of motoneuron subtypes as revealed by our PCA and is supported by previous work (*Nakanishi and Whelan, 2010*). A portion of this staggered rheobase current relates to differential maturation of passive properties, with delayed firing motoneurons increasing in size, as indexed by whole-cell capacitance, leading to a decrease in input resistance. This finding is in line with previous reports demonstrating increases in motoneuron physical size during the first two postnatal weeks (*Carrascal et al., 2005*; *Li et al., 2005*; *Vinay et al., 2000b*). However, previous work in adult cat motoneurons suggests that variation in cell size or input resistance cannot account for the full range of rheobase currents across a motor pool (*Fleshman et al., 1981*). Similarly, work in rat oculomotor neurons has demonstrated changes in rheobase during development that are linked to maturation of spike threshold and not changes in input resistance (*Carrascal et al., 2011*), suggesting that such observations are not species-specific. In addition to passive properties related to size, we found that active properties, determined by specific ion channels, differed between motoneuron subtypes and changed during development. This observation suggested that changes in active properties may also contribute to the rheobase separation of motoneuron subtypes from the second postnatal week onwards. These active properties produce nonlinearities in the input-output functions of motoneurons, based on distinct activation and inactivation properties of voltage-sensitive ion channels. The developmental emergence of a role for active properties in motoneuron recruitment may provide more flexible motor output than would be possible if recruitment was based on a fixed set of physical parameters such as motoneuron size. The integration of active properties may permit for more adaptable schemes of motoneuron recruitment as more complex motor behaviours emerge during later stages of postnatal development.

We first found that PICs differentially influence rheobase of motoneuron subtypes from the second postnatal week. PICs are well described in motoneurons (*Heckman et al., 2007*; *Hounsgaard et al., 1988*; *Hounsgaard and Kiehn, 1989*; *Li and Bennett, 2003*; *Quinlan et al., 2011*; *Schwindt and Crill, 1980*) where they amplify synaptic inputs to distal dendrites (*Carlin et al., 2009*; *Carlin et al., 2000*; *Heckman et al., 2008*) and are critical for the generation of repetitive firing (*Bouhadfane et al., 2013*; *Kuo et al., 2006*; *Miles et al., 2005*). In some cases, PICs also produce plateau potentials and lead to bistability, which can sustain repetitive firing following the termination of excitatory synaptic input (*Lee and Heckman, 1998b*). We expected to find larger PICs in slow motoneurons given their lower rheobase currents and greater tendency for bistability (*Lee and Heckman, 1998a*), which is thought to be important for the maintenance of posture. However, we found that PIC amplitude was larger in delayed compared to immediate firing motoneurons, although when correcting for cell size we found no difference in PIC density between subtypes. Further analyses revealed a difference in the activation voltages of PICs across motoneuron subtypes, with PICs activating at more depolarized potentials in delayed firing motoneurons. These data support that variations in the activation voltage of PICs help drive differential recruitment across motoneuron subtypes. Our findings are also

consistent with work in adult cats, which show that a more hyperpolarized PIC activation voltage in smaller motoneurons is a better predictor of bistability than PIC amplitude (*Lee and Heckman, 1998a*). While we rarely observed bistability or self-sustained firing, it is possible that the population of immediate firing motoneurons in our study may be more prone to bistability in the presence of neuromodulation, such as is included in other studies (*Hounsgaard et al., 1988*; *Hounsgaard and Kiehn, 1985*; *Lee and Heckman, 1998a*).

PICs are produced by a complement of ion channels that conduct sodium and calcium currents. Our pharmacological analysis demonstrated that Nav1.6 channels contribute to the PICs in both motoneuron subtypes, with an additional contribution of L-type calcium channels in delayed firing motoneurons. Although it is important to note that additional ion channels and their differential expression may contribute to the differences in properties of PICs in motoneuron subtypes (*Bouhadfane et al., 2013*). Notably, we find that blockade of Nav1.6 and L-type calcium channels only produced a combined 53% reduction in PIC amplitude in delayed firing motoneurons, and blockade of Nav1.6 produced a 60% reduction in PIC amplitude in immediate firing motoneurons. We found that Nav1.6, rather than L-type calcium channels, determines the PIC activation voltage and shapes rheobase of delayed firing motoneurons by controlling the voltage trajectory up to and including spike threshold. This is in line with previous experimental (*Kuo et al., 2006*) and computational work (*Zhang and Dai, 2020*). Nav1.6 channels are well-placed to control rheobase, given their dense expression at the AIS (*Brocard et al., 2016*; *Jørgensen et al., 2021*), a specialized structure that is critical for the generation of action potentials and support of repetitive firing. Physical properties of the AIS may also contribute to differential recruitment of motoneuron subtypes, given that the AIS is further from the soma in fast compared to slow motoneurons (*Rotterman et al., 2021*). If so, one might predict that the distance between soma and AIS may increase in fast motoneurons during the first two postnatal weeks and contribute to functional specification of motoneuron subtypes.

Interestingly, while blocking Nav1.6 reduced the overall PIC amplitude and depolarized spike threshold in immediate firing motoneurons, it did not alter the onset voltage of the PIC or change their rheobase, which is supportive of PIC onset voltage being more important for shaping rheobase. One possibility is that PIC onset and rheobase in immediate firing motoneurons are controlled by other ion channels (*Bouhadfane et al., 2013*). However, roles for calcium PICs in shaping motoneuron recruitment cannot be excluded in either motoneuron subtype and may have been underestimated in our transverse slice preparations due to a loss of key neuromodulatory sources that increase calcium PICs (*Conway et al., 1988*; *Hounsgaard et al., 1988*; *Hounsgaard and Kiehn, 1985*; *Li et al., 2007*), loss of dendrites in our slice preparations where L-type calcium channels are clustered (*Carlin et al., 2000*; *Elbasiouny et al., 2005*), and the nature of the approaches where recruitment was assessed; with current injected at the soma being more likely to activate Nav1.6 channels on the AIS before calcium channels of the dendrites.

Other channels, including Kv1.2 channels, which underlie the characteristic delayed firing behaviour of fast motoneurons (*Bos et al., 2018*; *Leroy et al., 2015*), are also clustered at the AIS and may dynamically interact with Nav1.6 channels to shape rheobase. We found delayed firing motoneurons at all ages studied; however, changes in the expression of the underlying Kv1.2 channels that produce this firing behaviour, and the compartmental distribution of these channels, may contribute to changes in recruitment of motoneuron subtypes. Furthermore, Nav1.6-conducting sodium PICs are opposed by outward-conducting M-type potassium currents mediated by Kv7 channels that are located on the AIS of excitatory spinal interneurons. Given that M-type currents can influence measurements of PICs in voltage-clamp mode and oppose the physiological roles of sodium PICs in the generation of locomotor rhythmogenesis (*Verneuil et al., 2020*), they could also influence motoneuron recruitment. Thus, we cannot rule out potential roles for potassium channels in the differential maturation of motoneuron subtype recruitment, given that these conductances were not blocked in our study.

We found that rheobase of delayed firing motoneurons increased further from the third postnatal week, paralleling the emergence of more complex locomotor behaviours, similar to those seen in adults (*Altman and Sudarshan, 1975*). However, this increase in delayed firing motoneuron rheobase could not be accounted for by changes in passive properties or PICs. Instead, we found subtype-specific changes in a hyperpolarization-activated inward current (Ih) that we hypothesized may contribute to this further refinement of delayed firing motoneuron rheobase. Ih is a nonspecific inward current conducted by HCN channels, which are located on the somata and proximal dendrites

of larger motoneurons in the rodent spinal cord (*Milligan et al., 2006*). Although depolarizing sag potentials (*Delestrée et al., 2014*; *Gustafsson and Pinter, 1984*; *Ito and Oshima, 1965*; *Kiehn et al., 2000*; *Manuel et al., 2009*; *Meehan et al., 2010*) and developmental changes in Ih have been reported previously in motoneurons (*Bayliss et al., 1994*; *Russier et al., 2003*; *Tsuzuki et al., 1995*), the functional significance of these findings was unclear. We found a prominent increase in Ih density, measured at voltages near resting membrane potential (–70 mV), in delayed but not immediate firing motoneurons and evidence of a significant h-current measured at the resting potential of delayed firing motoneurons at week 3. Blocking HCN channels with ZD7288 or ivabradine led to a decrease in rheobase of delayed but not immediate firing motoneurons when depolarizing ramps were initiated from a membrane potential of –70 mV. A high conductance state produced by a significant h-current that is active at resting potential could diminish the influence of excitatory synaptic input and delay the recruitment of fast motoneurons by acting as a shunting conductance. Recruitment may be further delayed due to the progressive loss of inward h-current during depolarization of the membrane potential. Importantly, the resting membrane potential is dynamic, and the influence of Ih on fast motoneuron recruitment will become diminished at more depolarized potentials. However, motoneurons are also often hyperpolarized below –70 mV during periods of inhibition, with fast motoneurons receiving a higher density of inhibitory synaptic inputs compared to slow motoneurons (*Allodi et al., 2021*). Therefore, the differential weighting of synaptic and intrinsic properties may be a key feature that contributes to the differential recruitment of motoneuron subtypes.

In addition to decreasing the rheobase of delayed firing motoneurons, both ZD7288 and ivabradine also hyperpolarized the resting membrane potential of both motoneuron subtypes, albeit to a much smaller extent in immediate compared to delayed firing motoneurons. These data suggest that Ih may contribute to the resting membrane potential of both motoneuron subtypes. However, hyperpolarization of the resting potential of immediate firing motoneurons by the two HCN channel blockers was perhaps unexpected given that we found no detectable h-current at voltages near resting membrane potential in this motoneuron subtype. It is possible that other currents that oppose Ih may occlude its clear measurement in immediate firing motoneurons (*Buskila et al., 2019*; *Kjaerulff and Kiehn, 2001*; *MacLean et al., 2003*; *Picton et al., 2018*) and that the contribution of Ih is only revealed following application of ZD7288 or ivabradine. In support of this possibility, we observed a slow hyperpolarization of the membrane potential during negative current steps in both immediate firing motoneurons at baseline and in delayed firing motoneurons in the presence of ZD7288 (*Figure 6A*). Alternatively, space clamp issues due to the large surface area and dendritic arbor of motoneurons may have led to the underestimation of h-currents. While it is expected that this error would have been greatest in the larger, delayed firing motoneurons, it is possible that this error might have been sufficient to mask the small but significant h-current in immediate firing motoneurons.

Overall, this further refinement of motoneuron subtype recruitment due to changes in Ih during the third postnatal week may not only support orderly recruitment but may also be important for decreasing the recruitment gain of the motoneuron pool to support complex motor behaviours that require a higher degree of precision compared to earlier developmental stages (*Figure 7*).

Our analysis provides novel insight into mechanisms that shape rheobase and may contribute to the orderly recruitment of motoneuron subtypes, extending our understanding of the mechanisms controlling the gradation of muscle force beyond the size principle. Motoneuron subtypes also differ in synaptic properties including the circuit of origin, synaptic weight, and neurotransmitter release dynamics (*Binder et al., 2002*; *Gabriel et al., 2011*; *McLean et al., 2007*; *Song et al., 2020*). For example, synaptic inputs originating from a variety of premotor sources may terminate on different motoneuron compartments that are associated with clusters of specific ion channels. Therefore, synapses that terminate on distal dendrites compared to more proximal loci, such as the proximal dendrites, soma, or AIS, may engage different voltage-sensitive ion channels leading to differential recruitment. It is likely that functionally specified motor pools may rely on varying degrees of synaptic and intrinsic properties to tune recruitment gain. Furthermore, differential control of intrinsic and synaptic properties of motoneuron subtypes by neuromodulators likely also adjusts recruitment gain within or across motor pools (*Bertuzzi and Ampatzis, 2018*; *Jha and Thirumalai, 2020*).

Given the new roles that we show here for PICs and Ih in the differential control of motoneuron subtypes, it is possible that neuromodulation of these currents (*Conway et al., 1988*; *Harvey et al., 2006b*; *Hounsgaard et al., 1988*; *Kjaerulff and Kiehn, 2001*; *Revill et al., 2019*; *Sirois and Bayliss,*

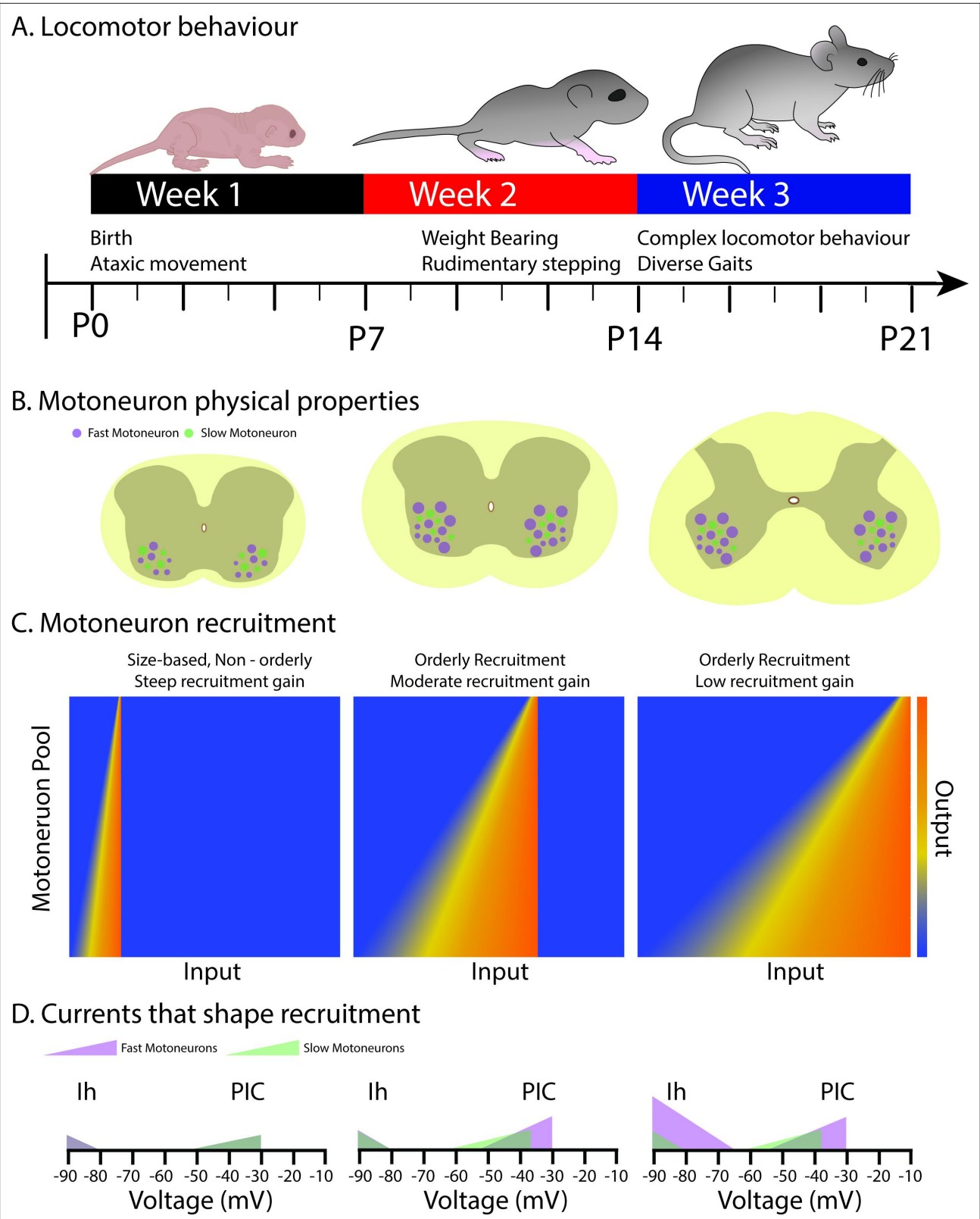

**Figure 7.** Schematic summarizing changes in the physical properties, recruitment, and underlying currents in motoneuron subtypes that shape the orderly recruitment and recruitment gain of motoneurons during postnatal development as complex locomotor behaviours emerge. (**A**) Illustrates the time course over which locomotor behaviours mature during the first three weeks of postnatal development (adapted from *Jean-Xavier et al., 2018*). (**B**) depicts the increase in the size of fast, but not slow motoneurons during the first two postnatal weeks, with no further increases into the third. (**C**)

*Figure 7 continued on next page*

*Figure 7 continued*

The range of inputs over which a motor pool is recruited expands across the first three weeks of postnatal development. (**D**) The expansion of the recruitment range coincides with the maturation of persistent (PIC) and hyperpolarization-activated (Ih) inward currents. Fast motoneurons and their underlying currents are depicted in purple and slow motoneurons, and their underlying currents depicted in green.

---

*2002*), as descending sources of neuromodulators reach the lumbar spinal cord during development (*Bregman, 1987*; *Commissiong, 1983*), may not only shape the emergence of orderly recruitment (*Pflieger et al., 2002*; *Smith and Brownstone, 2020*; *Vinay et al., 2005*; *Vinay et al., 2000a*; *Vinay et al., 2002*), but also provide a means to dynamically modulate recruitment gain.

We set out to identify key intrinsic properties, and the underlying currents, that shape motoneuron rheobase and aimed to determine when these currents are integrated during postnatal development to establish recruitment order amongst motoneuron subtypes. Our results support the notion that fast motoneurons are not simply scaled-up slow motoneurons (*Kernell and Zwaagstra, 1981*). While passive properties linked to motoneuron size (i.e., capacitance, input resistance) do help to stagger the recruitment of fast and slow motoneurons, we also reveal a key role for active properties, produced by PICs and Ih, which are sequentially integrated during the first three weeks of postnatal development in parallel with the emergence of increasingly complex motor behaviours. Our work will inform future studies aimed toward understanding mechanisms that enable the balance of motor control to shift between the generation of vigorous movements versus precision movements, as is needed during development and when adapting to different motor tasks.

## Materials and methods

### Animals

Experiments were performed on tissue obtained from 133 (male: n = 68; and female: n = 65) wild-type C57Bl/6J mice at postnatal days (P) 1–20. All procedures performed were conducted in accordance with the UK Animals (Scientific Procedures) Act 1986 and were approved by the University of St Andrews Animal Welfare Ethics Committee (PPL: P6F7B721E).

### Tissue preparation

All animals were sacrificed by performing a cervical dislocation followed by rapid decapitation. Animals were eviscerated and pinned ventral side up in a dissecting chamber lined with silicone elastomer (Sylguard) filled with carbogenated (95% oxygen, 5% carbon dioxide), ice-cold (1–2°C) potassium gluconate-based dissecting/slicing aCSF (containing in mM: 130 K-gluconate, 15 KCl, 0.05 EGTA, 20 HEPES, 25 D-glucose, 3 kynurenic acid, 2 Na-pyruvate, 3 myo-inositol, 1 Na-L-ascorbate; pH 7.4, adjusted with NaOH; osmolarity approximately 345 mOsm). Spinal cords were exposed by performing a ventral laminectomy, cutting the ventral roots and gently lifting the spinal cord from the spinal column. Spinal cords were removed within 3–5 min following cervical dislocation. Spinal cords were secured directly to an agar block (3% agar) with VetBond surgical glue (3 M) and glued to the base of the slicing chamber with cyanoacrylate adhesive. The tissue was immersed in pre-carbogenated ice-cold dissecting/slicing aCSF. Blocks of frozen slicing solution were also placed in the slicing chamber to keep the solution around 1–2°C. On average, the first slice was obtained within 10 min of decapitation, which increased the likelihood of obtaining viable motoneurons in slices. 300 µm transverse slices were cut at a speed of 10 µm/s on the vibratome (Leica VT1200) to minimize tissue compression during slicing. 3–4 slices were obtained from each animal. Slices were transferred to a recovery chamber filled with carbogenated pre-warmed (35°C) recovery aCSF (containing in mM: 119 NaCl, 1.9 KCl, 1.2 NaH$_2$PO$_4$, 10 MgSO$_4$, 1 CaCl, 26 NaHCO$_3$, 20 glucose, 1.5 kynurenic acid, 3% dextran) for 30 min after completion of the last slice which took 10–15 min on average. Following recovery, slices were transferred to a chamber filled with carbogenated warm (35°C) recording aCSF (containing in mM: 127 NaCl, 3 KCl, 1.25 NaH$_2$PO$_4$, 1 MgCl, 2 CaCl$_2$, 26 NaHCO$_3$, 10 glucose) and allowed to equilibrate at room temperature (maintained at 23–25°C) for at least 1 hr before experiments were initiated.

---

## Whole-cell patch-clamp electrophysiology

Whole-cell patch-clamp recordings were obtained from 345 lumbar motoneurons. We typically recorded from 2 cells (mode), with a range of 1–6 cells per animal. Slices were stabilized in a recording chamber with fine fibres secured to a platinum harp and visualized with a ×40 objective with infrared illumination and differential interference contrast. A large proportion of the motoneurons studied were identified based on location in the ventrolateral region with somata greater than 20 µm. Recordings were obtained from a subset of motoneurons that had been retrogradely labelled with Fluoro-Gold (Fluorochrome, Denver, CO). Fluoro-Gold was dissolved in sterile saline solution and 0.04 mg/g injected intraperitoneally 24–48 hr prior to experiments (*Miles et al., 2005*). In addition to recording from larger FG-positive cells, this approach allowed us to more confidently target smaller motoneurons. Motoneurons were visualized and whole-cell recordings obtained under DIC illumination with pipettes (L: 100 mm, OD: 1.5 mm, ID: 0.84 mm; World Precision Instruments) pulled on a Flaming Brown micropipette puller (Sutter instruments P97) to a resistance of 2.5–3.5 MΩ. Pipettes were back-filled with intracellular solution (containing in mM: 140 KMeSO$_4$, 10 NaCl, 1 CaCl$_2$, 10 HEPES, 1 EGTA, 3 Mg-ATP, and 0.4 GTP-Na2; pH 7.2–7.3, adjusted with KOH).

Signals were amplified and filtered (6 kHz low-pass Bessel filter) with a Multiclamp 700 B amplifier, acquired at 20 kHz using a Digidata 1440A digitizer with pClamp version 10.7 software (Molecular Devices) and stored on a computer for offline analysis.

## Identification of motoneuron subtypes

Motoneuron subtypes were identified using a protocol established by *Leroy et al., 2014* and *Leroy et al., 2015*, which differentiates motoneuron type based on the latency to the first spike when injecting a 5 s square depolarizing current near the threshold for repetitive firing. Using this approach, we were able to identify two main firing profiles: a delayed repetitive firing profile with accelerating spike frequency, characteristic of fast-type motoneurons, and an immediate firing profile with little change in spike frequency, characteristic of slow-type motoneurons (*Figure 1*; *Leroy et al., 2015*; *Leroy et al., 2014*). Delayed and immediate firing motoneurons were identified in equivalent proportions at each developmental time point studied (week 1: delayed: 58%; immediate: 42%; week 2: delayed: 67%; immediate: 33%; week 3: delayed: 65%; immediate: 35%) and were found in rostral and caudal segments of male and female mice. There were no apparent differences in the relative proportions of the two motoneuron types when comparing spinal segment or animal sex. Data analysis was performed whilst blinded to motoneuron subtype.

All motoneuron intrinsic properties were studied by applying a bias current to maintain the membrane potential at –60 mV, or at –70 mV for the subset of experiments with Ih or PIC blockers. Values reported are not liquid junction potential corrected to facilitate comparisons with previously published data (*Durand et al., 2015*; *Miles et al., 2007*; *Nascimento et al., 2020*; *Nascimento et al., 2019*; *Quinlan et al., 2011*; *Smith and Brownstone, 2020*). Cells were excluded from analysis if access resistance was greater than 20 MΩ or changed by more than 5 MΩ over the duration of the recording, if resting membrane potential deviated by more than 5 mV over the duration of the recording period, or if spike amplitude measured from threshold (described below) was less than 60 mV.

## Pharmacology

We deployed pharmacological tools to probe the contribution of ion channels contributing to PICs and hyperpolarization activated inward currents (Ih). Nifedipine (20 µM; Tocris) was used to assess the contribution of L-type calcium channels, and 4,9-AH-TTX (200 nM; Tocris) to assess the contribution of Nav1.6 channels to PIC and recruitment. Ih was blocked with the selective HCN channel blocker, ZD7288 (10 µM; Tocris) or ivabradine (10 µM; Tocris).

## Data acquisition and analysis

Passive properties including capacitance, membrane time constant (tau), and passive input resistance (Ri) were measured during a hyperpolarizing current steps that brought the membrane potential from –60 to –70 mV. Input resistance was measured from the initial voltage trough to minimize the impact of slower-acting active conductances (e.g., Ih, sag). The time constant was measured as the time it took to reach 2/3 of the peak voltage change. Capacitance was calculated by dividing the time constant

by the input resistance (C = T/R). Resting membrane potential was measured 10 min after obtaining a whole-cell configuration and at the end of recordings from the MultiClamp Commander.

Rheobase and repetitive firing properties were assessed during slow (100 pA/s) depolarizing current ramps, which allowed us to measure the rheobase current and voltage threshold of the first action potential. Rheobase was also measured during long (5 s) depolarizing current steps that were used to determine the motoneuron firing type (delayed or immediate) and defined as the first current step to produce at least three action potentials. Recruitment gain was qualitatively illustrated by generating cumulative proportion histograms of recruitment current values across respective motoneuron samples at developmental time points or before and after drug application. The voltage threshold of the first action potential was defined as when the change in voltage reached 10 mV/ms. Firing rates were measured at rheobase, 2× rheobase, and derecruitment (maximum firing rate). As previously reported, motoneurons displayed sub-primary and primary regions of the FI relationship. Sub-primary range gain was measured from the slope of a linear function fitted to the initial section of the FI relationship (*Jensen et al., 2018*; *Manuel et al., 2009*; *Meehan et al., 2010*). We did not find evidence for a secondary firing range in the motoneurons studied.

The subthreshold trajectory of the membrane potential measured in response to a slow depolarizing current ramp is composed of passive and active elements defined by linear and exponential changes in membrane potential, respectively (*Delestrée et al., 2014*; *Jørgensen et al., 2021*; *Kuo et al., 2006*). The slope of this linear phase was calculated as a proxy for passive excitability. The onset of the active, exponential component of the membrane depolarization was defined as the point at which the membrane potential deviated by more than 1 mV from the linear function of the passive phase. The current and voltage values at the onset of the voltage acceleration were compared to the rheobase current and voltage threshold measured from the first action potential elicited on the depolarizing current ramp. A hypothetical 'passive rheobase' current was then estimated by extrapolating the line applied to the passive depolarizing phase and measuring its intersection with the voltage threshold of the first spike. The difference between this hypothetical passive rheobase current and the measured rheobase current was calculated to quantify the influence that active conductances (presumably PIC) have on recruitment.

The influence that PICs exert on motoneuron excitability was also estimated in current-clamp mode using the injection of a triangular, depolarizing current ramp with 5 s rise and fall times (*Durand et al., 2015*; *Leroy et al., 2015*; *Li and Bennett, 2003*; *Steele et al., 2020*). Depolarizing current ramps were set to a peak current of 2× repetitive firing threshold current (determined with a 100 pA/s depolarizing current ramp initiated from –60 mV). The influence of PICs on motoneuron excitability was estimated by calculating the difference (Delta I) between the current at firing onset on the ascending component of the ramp and the current at derecruitment on the descending component of the ramp. A negative Delta I is suggestive of a PIC. Firing hysteresis was also assessed on the ascending and descending components of the ramp. We subdivided cells into one of four types based on the firing hysteresis on ascending and descending portions of the ramp (*Bennett et al., 1998*; *Cotel et al., 2009*; *Durand et al., 2015*; *Li and Bennett, 2003*) (type 1: linear; type 2: adapting; type 3: sustained; type 4: counterclockwise).

PICs were measured in voltage clamp by injecting a slow depolarizing voltage ramp (10 mV/s from –90 to –10 mV) over 8 s (*Huh et al., 2021*; *Quinlan et al., 2011*; *Steele et al., 2020*). PIC onset voltage, voltage at which the peak current occurs (voltage peak), peak current amplitude, and peak current density were measured from post hoc leak-subtracted traces as in *Quinlan et al., 2011*, *Steele et al., 2020*, and *Verneuil et al., 2020*.

Input resistance was calculated by taking the slope of the current-voltage relationship at initial (peak voltage trough at the start) and steady-state (plateau at end) voltage deflection during a series of 1 s hyperpolarizing current steps. Sag conductances were estimated as the difference in conductance (1/R) measured during initial and steady-state voltage plateau at the end of hyperpolarizing current steps (($1/Rin_{ss}$) – ($1/Rin_{initial}$)). Ih was measured in voltage clamp during 1 s hyperpolarizing voltage steps from –60 mV to –110 mV, in 10 mV increments. A resting Ih was also measured by stepping the voltage down from a holding potential of –50 mV to the respective resting membrane potential of a given cell in a subset of motoneurons. Ih was measured as the difference between the initial and steady-state current within similar windows analysed for depolarizing sag in current clamp. Ih time constant (tau) was measured as the time (in ms) to reach 2/3 peak Ih at each respective voltage step.

Single spikes and AHPs were elicited using a 10 ms square, depolarizing current pulse applied at an intensity 25% above rheobase current. Spike threshold was determined as the potential at which the derivative (dV/dt) increased above 10 mV/ms. Spike amplitude, rise time, and half width were measured from this point. mAHP properties (amplitude and half width) were measured from baseline (–60 mV). Single spike and mAHP properties were measured using event detection features in Clampfit version 11 (Molecular Devices).

### Research design and statistical analysis

Two-factor analysis of variance (ANOVA) was performed with motoneuron type (delayed and immediate) and postnatal week (weeks 1–3) as factors to address three core questions for each parameter studied: (1) Was there a difference between subtypes? (2) Did it change during development? (3) Do the subtypes mature differently during development? Statistical results addressing these questions are listed sequentially in *Supplementary file 1*: (1) main effect of subtype; (2) main effect of development; and (3): Subtype × development interaction. Holm–Sidak post hoc analysis was performed when significant main effects or interactions were detected with a significance level of p<0.05 and are summarized in *Supplementary file 1*. Given that these data are continuous and parametric, we performed Pearson correlations between rheobase current and passive properties (*Gustafsson and Pinter, 1984*) within each motoneuron subtype at each week. A simple linear regression, comparing the slopes of delayed and immediate motoneurons, was used to determine the degree of similarities or differences for each of the analyses performed (*Figure 3*). Two-factor repeated-measures ANOVA were conducted to test the effect of pharmacological agents on intrinsic properties and currents with MN type and drug as factors. Appropriate and equivalent nonparametric tests (Mann–Whitney or Kruskal–Wallis) were conducted when data failed tests of normality or equal variance with Shapiro–Wilk and Brown–Forsythe tests, respectively. Individual data points for all cells are presented in figures with mean ± SD. Statistical analyses were performed using GraphPad version 9.0 (Prism, San Diego, CA, USA).

PCA was performed using GraphPad version 9.0 to determine global changes in motoneuron intrinsic properties during the first three weeks of postnatal development. 25 intrinsic properties from 261 motoneurons across weeks 1–3 were included in the analysis. PC scores (PC Scores) for each cell were extracted for each PC, and changes between subtypes during development were analysed using a two-factor ANOVA. 3D scatterplots between PC scores (PCs 1–3) were generated in Sigmaplot 14.0 (Sigmastat, San Jose, CA).

## Acknowledgements

We are grateful for the guidance received from Dr. Marco Beato and Dr. Filipe Nascimento on approaches to prepare lumbar slices from functionally mature mice and Frankie Sorrell for comments on an earlier version of the manuscript.

## Additional information

### Funding

| Funder | Grant reference number | Author |
| --- | --- | --- |
| Royal Society | NIF/R1/180091 | Simon A Sharples |
| Natural Sciences and Engineering Research Council of Canada | NSERC-PDF-517295-2018 | Simon A Sharples |

The funders had no role in study design, data collection and interpretation, or the decision to submit the work for publication.

## Author contributions

Simon A Sharples, Conceptualization, Data curation, Formal analysis, Funding acquisition, Investigation, Methodology, Writing – original draft, Writing – review and editing; Gareth B Miles, Conceptualization, Funding acquisition, Resources, Supervision, Writing – review and editing

## Author ORCIDs

Simon A Sharples ⓘ http://orcid.org/0000-0003-2316-1504
Gareth B Miles ⓘ http://orcid.org/0000-0002-8624-4625

## Ethics

All procedures performed were conducted in accordance with the UK Animals (Scientific Procedures) Act 1986 and were approved by the University of St Andrews Animal Welfare Ethics Committee. (PPL: P6F7B721E).

## Decision letter and Author response

Decision letter https://doi.org/10.7554/eLife.71385.sa1
Author response https://doi.org/10.7554/eLife.71385.sa2

## Additional files

### Supplementary files

• Supplementary file 1. Passive, recruitment, repetitive firing, and single spike properties of delayed and immediate firing motoneuron subtypes across three postnatal weeks. Number of motoneurons (n) included in each subtype (Del: delayed; Imm: immediate) across postnatal weeks 1–3. Passive properties include whole-cell capacitance, tau, input resistance, and resting membrane potential (RMP). Recruitment properties: rheobase, derecruitment current, current firing range, voltage threshold of the first spike, and relative distance between RMP and spike threshold. Repetitive firing properties measured following recruitment on slow depolarizing current ramps include minimum firing rate, firing rate at 2× rheobase, maximum firing rate, and the slope of the sub-primary range (SPR). Single spike (action potential) properties including spike amplitude, rise time, and half width were measured from threshold, and medium afterhyperpolarization (mAHP) amplitude and half width, were elicited with a 10 ms square current pulse applied at 1.25× rheobase. Statistics: data were analysed using two-factor ANOVA with MN type (Del and Imm) and developmental week (weeks 1–3) as factors. Statistical analyses addressed three core questions: (1) Was there a difference between subtypes? (2) Did it change during development? (3) Do the subtypes mature differentially during development? Statistical results listing F and p values from two-factor ANOVA addressing these questions are listed sequentially: (1) main effect of subtype; (2) main effect of development; and (3) subtype × development interaction. All data are presented as mean ± SD (min, max). p-Values are derived from Holm–Sidak post hoc comparisons within weeks, between MN types. Superscript numbers denote significant differences from Holm–Sidak within MN types, between weeks (1: week 1; 2: week 2; 3: week 3).

• Supplementary file 2. Active properties of delayed and immediate firing motoneuron subtypes across three postnatal weeks. Characteristics of the subthreshold membrane potential trajectory during slow depolarizing recruitment current ramps. Presented are the acceleration onset voltage, acceleration amplitude, passive recruitment current estimated based on the initial linear trajectory of the membrane potential, actual measured recruitment current, and an estimate of the underlying persistent inward current (ePIC) that produces the membrane potential acceleration. Delta I reflects the difference in derecruitment and recruitment currents on triangular depolarizing current ramps and provides an additional estimate of PIC. PICs were measured in voltage clamp during weeks 2–3 with the onset voltage, peak voltage, current amplitude, and densities measured. Sag conductance was estimated during weeks 1–3 and underlying h current (Ih) measured with amplitude and density at –70 mV and –110 mV presented. Time constant (tau) of Ih. Statistics: data were analysed using two-factor ANOVA with MN type (delayed and immediate) and developmental week (weeks 1–3) as factors. Statistical analyses addressed three core questions: (1) Was there a difference between subtypes? (2) Did it change during development? (3) Do the subtypes mature differentially during development? Statistical results listing F and p values from two-factor ANOVA addressing these questions are listed sequentially: (1) main effect of subtype; (2) main effect of development; (3) subtype × development interaction. All data are presented as mean ± SD (min, max). Number of cells (n) are included in brackets for each group. p-Values are derived from Holm–Sidak post hoc comparisons within weeks, between MN types. Superscript numbers denote significant differences

from Holm–Sidak within MN types, between weeks (1: week 1; 2: week 2; 3: week 3).

• Supplementary file 3. A comparison of studies on lumbar motoneuron intrinsic properties recorded with whole-cell patch-clamp electrophysiology across the first three weeks of postnatal development in mice. Comparison of intrinsic properties measured in studies on motoneuron subtypes (current work; *Leroy et al., 2014*) and those that pooled all lumbar motoneurons studied (*Quinlan et al., 2011*; *Nakanishi and Whelan, 2010*; *Smith and Brownstone, 2020*). Data are presented as mean ± SD; *Nakanishi and Whelan, 2010* are presented as mean ± SEM.

• Transparent reporting form

• Source data 1. Values for intrinsic properties studied in motoneuron subtypes across the first, second, and third postnatal weeks and are summarized in *Supplementary file 1*.

• Source data 2. Values for active properties studied in motoneuron subtypes across the first, second, and third postnatal weeks and are summarized in *Supplementary file 2*.

### Data availability

Source data included in all figures (Figures 2-6 + Figure 3 – Figure Supplement 1, Figure 4 – Figure Suuplement 1), and Supplementary File 1 and 2 have been uploaded as excel files with the manuscript.

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
