## [Editor Report]

This manuscript will be of interest to those studying the neuroscience of movement as it addresses a fundamental aspect of movement: motoneuron recruitment. The authors use spinal cord slices to provide a comprehensive analysis of motoneuron intrinsic properties, passive and active, that mature in the early postnatal period in mice and coincide with their differentiation into ‘slow’ and ‘fast’ phenotypes. They argue that these properties together shape motoneuron recruitment, suggesting that textbook views of orderly recruitment may be oversimplifications.

---

## [Decision Letter]

**Decision letter after peer review:**

Thank you for submitting your article "Maturation of persistent and hyperpolarization-activated inward currents shape the recruitment of motoneuron subtypes during postnatal development" for consideration by *eLife*. Your article has been reviewed by 4 peer reviewers, including Rob Brownstone as Reviewing Editor and Reviewer #1, and the evaluation has been overseen by John Huguenard as the Senior Editor. The following individual involved in review of your submission has agreed to reveal their identity: Claire Meehan (Reviewer #2).

Essential revisions:

All four reviewers agree that this work is very well done and that an impressive amount of data has been presented. The work will be of interest to a broad range of neuroscientists, including those studying the neuroscience of movement and also those studying how populations of neurons with variable properties may be recruited in circuits. However, a few important points have been raised by the reviewers that should help the authors to revise their manuscript. As written, we are concerned about over-interpretation of the data.

1. Motoneuron typing. The authors relay on surrogate markers of motoneuron type – delayed vs immediate firing. They then use ramp currents (e.g. Figure 3A) rather than the established rectangular pulses to differentiate the two. But the slow ramp depolarisation used to measure rheobase may lead to inactivation of some currents. Furthermore, motoneuron types are based on the type of muscle fibre that motor neuron innervates, and this has not been directly established for the recorded motoneurons. Given the above, it would be prudent to primarily use the terms delayed and immediate firing rather than fast and slow until the Discussion.

2. The term "recruitment" is interesting. The authors are using intracellular current injection to study rheobase. The degree to which this reflects recruitment during behaviour or even via "evenly" spread inputs (such as Ia afferents) across the motor pool is not clear. For example, dendritic inputs may recruit active currents that are not as readily recruited during somatic current injection. While the findings have implications for considering recruitment, the focus on recruitment per se should be toned down as the relation is only possible, not definite. On this note, we would suggest that even the title be changed, e.g. to "activation" or "activation properties," for example.

3. PICs. The authors suggest that recruitment of PICs is related to motoneuron recruitment, but the pharmacological experiments are reported in only delayed firing motoneurons. These experiments should be extended to immediate firing motoneurons.

4. There seem to be some discrepancies in the Ih data as outlined in the reviews. For example, the voltage clamp data do not show Ih above -70mV, yet ZD seems to have an effect at resting potential in current clamp experiments. If Ih is not active at rest, how does it contribute to rheobase/recruitment? These points should be clarified, possibly through additional data.

*Reviewer #1 (Recommendations for the authors):*

– Significant digits are generally good, but should be checked: e.g. 14.02% on p.3.

– ePIC abbreviation – why the abbreviation, which is also unclear?

– Some statements are made without references and should be checked, e.g. line 240ish.

– Line 474 – fine motor control? Not at all clear, seems like a bit of a leap.

– Stats – well done. One point: seems like experimental unit was the MN and not the mouse, which is okay (lots of mice used) but should be explicit and justified, with an indication of mode (and range) number of MNs recorded per mouse.

– Not all figure panels are called out in the text (starting with Figure 1A for example).

– It's not clear that the Pearson correlation coefficient is the correct statistic, rather than the Spearman.

*Reviewer #2 (Recommendations for the authors):*

I must start this by stating what pleasure it was to read this manuscript. In fact, it is one of the best manuscripts I have read in long time. It is well written, clear and concise with data presented in a clear and transparent way clearly supporting the conclusions. I consequently only have only relatively minor suggestions as I think the manuscript reads extremely well as is and I am reluctant to suggest changes for the sake of it.

*Reviewer #3 (Recommendations for the authors):*

Could you confirm some of the results using a different way to identify motoneuron type? For example, using retrograde labelling of motoneurons innervating muscles that are primarily Fast or slow type.

Considering that PICs are sensitive to neuromodulators, is there any possibility that the lack of neuromodulators in your patch-clamp experiments could account for the differences in PICs between the cell types that you observed?

*Reviewer #4 (Recommendations for the authors):*

1) It is advisable to measure Ih by a ZD7288-sensitive current obtained by subtraction.

2) An original study (Leroy et al., 2014) previously showed in neonatal mice that the input conductance is smaller, the rheobase is lower and the voltage threshold for spiking is more hyperpolarized in immediate firing motoneurons (slow MNs) than in delayed firing ones (fast MNs) during the 2nd postnatal week. Furthermore, the study reported that delayed firing motoneurons have larger soma size and dendritic length than immediate firing motoneurons. The results of this study should be clearly stated in the discussion.

3) In cats, the threshold of persistent inward currents has already been reported to be more hyperpolarized in small MNs and can be referred (Lee and Heckman, 1998).

---

## [Author Response]

Essential revisions:All four reviewers agree that this work is very well done and that an impressive amount of data has been presented. The work will be of interest to a broad range of neuroscientists, including those studying the neuroscience of movement and also those studying how populations of neurons with variable properties may be recruited in circuits. However, a few important points have been raised by the reviewers that should help the authors to revise their manuscript. As written, we are concerned about over-interpretation of the data.1. Motoneuron typing. The authors relay on surrogate markers of motoneuron type – delayed vs immediate firing. They then use ramp currents (e.g. Figure 3A) rather than the established rectangular pulses to differentiate the two. But the slow ramp depolarisation used to measure rheobase may lead to inactivation of some currents. Furthermore, motoneuron types are based on the type of muscle fibre that motor neuron innervates, and this has not been directly established for the recorded motoneurons. Given the above, it would be prudent to primarily use the terms delayed and immediate firing rather than fast and slow until the Discussion.

We have changed all reference to fast and slow motoneurons within the methods and Results sections to delayed and immediate firing motoneurons and included a section in our discussion highlighting the merits of this approach and linking these surrogate electrophysiological markers to those of different motor unit types in the second paragraph of the discussion (Page 16 Line 516-528).

Perhaps we were not fully clear in our description of protocols used to identify motoneuron subtype and those used to measure rheobase. Motoneuron subtype was always determined using the established 5 second square depolarizing current steps. Depolarizing current ramps were only used to measure rheobase and not to identify motoneuron subtypes.

We used slow depolarizing current ramps to determine rheobase and spike threshold, for two reasons. First, the continuous nature of the ramp provides a high degree of resolution to measure rheobase, whereas traditional square steps are limited to the current intervals deployed by the experimenter, which for studies of motoneurons are often 50-100 pA apart. Second, we also chose to use current ramps to resemble those used to study recruitment of human motor units, where recruitment and rate coding of motor units have been extensively studied during ramp-stype muscle contraction paradigms (for examples, see Deluca et al., 1981; Oya et al., 2009; Del Vecchio et al., 2019). Indeed, as highlighted by the reviewers, voltage responses to square and ramp current injections likely involve disparate conductances due to temporal and voltage inactivation (or activation) properties of the ion channels involved, which could lead to differences in measured rheobase between protocols or across development. This is an interesting point, and likely speaks to the task dependence of motoneuron and motor unit recruitment. We agree that the cellular mechanisms that underlie task-dependence of recruitment is an interesting and exciting direction for future study.

We have now included additional analysis on rheobase currents during the 5s square depolarizing current step to supplement our data using the slow depolarizing ramp currents. Interestingly, and consistent with that reported by Leroy et al., (2014), the rheobase assessed on depolarizing ramps and that assessed with a 5 second square step to determine motoneuron subtype are strongly correlated at all ages studied (W1: r = 0.87 p < 1.0 e-15; W2: r = 0.95 p < 1.0 e-15; W3: r = 0.92 p < 1.0 e-15). Further, similar developmental changes in the rheobase of subtypes were found when assessed with a current step as those found with depolarizing current ramps. We have included these data as an additional Supplementary Figure (Figure 3 —figure supplement 1) and this result included in the Results section on Page 6, Line 173-178.

2. The term "recruitment" is interesting. The authors are using intracellular current injection to study rheobase. The degree to which this reflects recruitment during behaviour or even via "evenly" spread inputs (such as Ia afferents) across the motor pool is not clear. For example, dendritic inputs may recruit active currents that are not as readily recruited during somatic current injection. While the findings have implications for considering recruitment, the focus on recruitment per se should be toned down as the relation is only possible, not definite. On this note, we would suggest that even the title be changed, e.g. to "activation" or "activation properties," for example.

We have changed all references to recruitment within our methods and results to ‘rheobase’. We have also changed the title to ‘activation properties’ and included a section in our introduction (Line 68), results (Line 173-176) and discussion (Line 501) to better define the link between our measures of rheobase/activation current and reference to recruitment. Limitations on activation of motoneurons through somatic injection of current compared to more physiological activation are further discussed, with particular reference to potential roles for calcium PIC mediated by dendritic calcium channels. This discussion can be found on Page 19, Lines 620-626.

3. PICs. The authors suggest that recruitment of PICs is related to motoneuron recruitment, but the pharmacological experiments are reported in only delayed firing motoneurons. These experiments should be extended to immediate firing motoneurons.

We initially focused on the roles of PICs in shaping recruitment of delayed firing motoneurons during weeks 2 and 3 because we were trying to account for changes in rheobase that occur within delayed firing motoneurons during this period of postnatal development. However, our results cannot rule out a role for PICs in shaping immediate firing motoneuron rheobase as well. As suggested, we have therefore now performed an additional series of experiments with NaV1.6 and L-type calcium channel blockers on immediate firing motoneurons. These new data have been integrated into Figure 4 (Page 9, Lines 295-318), and results discussed (Page 19, Lines 616-620).

4. There seem to be some discrepancies in the Ih data as outlined in the reviews. For example, the voltage clamp data do not show Ih above -70mV, yet ZD seems to have an effect at resting potential in current clamp experiments. If Ih is not active at rest, how does it contribute to rheobase/recruitment? These points should be clarified, possibly through additional data.

We thank the reviewers for their careful assessment of these data. Our measurements of Ih in voltage clamp were initiated from holding potential of -60 mV, with the first step coming down to -70 mV. We have revised Figure 5 to better highlight the magnitude of the h-currents that we measured at -70 mV, which may have not been fully realized in the previous version of the figure that presented currents from the full IV that measured Ih at voltages ranging from -70 mV to -110 mV, with Ih increasing almost 10-fold when comparing currents measured at -70 mV and those measured at -110 mV. However, we recognize the resting potential of many of the neurons we studied is slightly more depolarized than -70 mV, and this was a key gap in our measurement of Ih. To address this issue, we have performed an additional set of recordings in delayed (n = 13) and immediate (n = 11) firing motoneurons from animals during week 3, taking measurements of Ih from a holding potential of -50 mV, stepping down to the respective resting membrane potential for each motoneuron studied. In line with previous measurements of Ih at -70 mV, we find a prominent Ih measured at resting potential in delayed but not immediate firing motoneurons. These data have been included in the Results section on Page 13, Line 429-437.

As pointed out by Reviewer 4, it is interesting that blocking HCN channels led to a hyperpolarization of the resting potential of the immediate firing motoneurons, even though we did not detect a measurable Ih at voltages near resting potential in this subtype. We have performed an additional set of experiments with a second blocker of HCN channels (Ivabradine), which also produced a hyperpolarization of the RMP in both delayed and immediate firing motoneurons. Given that there are many currents that oppose Ih and its effects on membrane potential (eg. Kjaerulff and Kiehn, 2001; MacLean et al., 2003; 2005; Picton et al., 2018; Buskila et al., 2019), it is possible that these opposing currents might have masked a small, albeit significant, Ih in immediate firing motoneurons that was revealed following blockade of HCN channels with ZD7288 or ivabradine. In support of this notion, we also find a slow hyperpolarization of the membrane potential during negative current steps in immediate firing motoneurons at baseline and in delayed firing motoneurons in the presence of ZD7288. A trace illustrating this feature has been included in the revised Figure 6 and highlighted in the discussion on Page 20, Lines 673-675.

Alternatively, it is possible that space clamp issues, due to the large surface area, and extensive dendritic tree of motoneurons, may have led to an underestimate of the true magnitude of Ih currents measured in voltage clamp. While it is expected that this error might have been greatest in the larger, delayed firing motoneurons, it is possible that these Ih currents were sufficiently large in delayed firing motoneurons to be detected, whereas in the immediate firing motoneurons it was not. We have highlighted these caveats and possibilities in our discussion and can be found on Lines 675-679.

We have also expanded our discussion to highlight potential roles for Ih in shaping recruitment of delayed firing motoneurons during periods of inhibition, such as during rhythmic activity, where the membrane potential often dips below -70 mV. Fast fatigable (MMP9+) motoneurons have been shown to receive a greater density of inhibitory synaptic inputs, particularly those derived from V1 interneurons, compared to slow (ERRB+) motoneurons (Allodi et al., 2021), and this differential synaptic weighting may create greater opportunity for Ih to be engaged and contribute to staggering recruitment as our pharmacology data suggests. This discussion can be found on Lines 657-663.

Reviewer #1 (Recommendations for the authors):– Significant digits are generally good, but should be checked: e.g. 14.02% on p.3.

Significant digits have been checked and corrected.

– ePIC abbreviation – why the abbreviation, which is also unclear?

ePIC corresponds to estimated PIC, a phrase that is commonly used by those studying roles of PICs in the control of human motor units. It was our aim to use terminology that is commonly deployed in both human and animal studies. We have clarified this abbreviation when it is first mentioned in the Results section.

– Some statements are made without references and should be checked, e.g. line 240ish.

References have been added.

– Line 474 – fine motor control? Not at all clear, seems like a bit of a leap.

We were attempting to comment on the adaptability, potentially through neuromodulation, that might act to adjust properties linked to motoneuron recruitment and highlighting that physical properties are relatively static compared to voltage dependent active properties. We have expanded upon this statement, now on Page 18 lines 576-578, to clarify.

– Stats – well done. One point: seems like experimental unit was the MN and not the mouse, which is okay (lots of mice used) but should be explicit and justified, with an indication of mode (and range) number of MNs recorded per mouse.

The mode number of cells studied per animal was 2 (range 1-6). We have included the mode and range number of cells studied per mouse in the Methods section on Page 22, Line 758-759.

– Not all figure panels are called out in the text (starting with Figure 1A for example).

We have gone through the manuscript and ensured that all figure panels are cited in text.

– It's not clear that the Pearson correlation coefficient is the correct statistic, rather than the Spearman.

Given that our data are continuous and parametric, we performed Pearson Correlations, whereas a Spearman would be more appropriate for data that are categorical or non-parametric. This has been clarified in our statistics section on page 25, Line 885.

Reviewer #2 (Recommendations for the authors):I must start this by stating what pleasure it was to read this manuscript. In fact, it is one of the best manuscripts I have read in long time. It is well written, clear and concise with data presented in a clear and transparent way clearly supporting the conclusions. I consequently only have only relatively minor suggestions as I think the manuscript reads extremely well as is and I am reluctant to suggest changes for the sake of it.

Thank you for the kind words. We also really enjoyed performing these new experiments and preparing the manuscript as the results came together.

Reviewer #3 (Recommendations for the authors):Could you confirm some of the results using a different way to identify motoneuron type? For example, using retrograde labelling of motoneurons innervating muscles that are primarily Fast or slow type.

Whilst we agree that retrograde labelling would provide another useful way to identify motoneuron subtypes, we believe that this would be a considerable undertaking that is beyond the scope of the current work. Furthermore, as the reviewer points out, one could only target muscles that are primarily composed of fast or slow twitch fibres, precluding the definitive identification of either motoneuron subtype. As recommended in earlier reviewer comments, we have now toned down our definition of motoneuron subtypes to refer to delayed and immediate firing motoneurons, rather than fast and slow motoneurons. We have also included more discussion regarding the previously established relationship between delayed and immediate firing and putative fast and slow motoneuron subtypes (Page 16 Lines 516-528). In addition, we have included a supplementary table (Supplementary File 3) to highlight the similarities in intrinsic properties of the motoneurons included in our study with those reported in previous studies that have utilised molecular makers for fast and slow motoneurons (MMP9, Chondrolectin, and ERRB).

Considering that PICs are sensitive to neuromodulators, is there any possibility that the lack of neuromodulators in your patch-clamp experiments could account for the differences in PICs between the cell types that you observed?

This is an excellent point; differential neuromodulation of PICs could represent an important additional factor that differentiates motoneuron subtypes. We have now acknowledged this limitation in our discussion on roles for PICs in motoneuron recruitment. A discussion of such caveats can be found on Page 19, Line 620-626. Modulation of properties related to recruitment is indeed an exciting area of ongoing study.

Reviewer #4 (Recommendations for the authors):1) It is advisable to measure Ih by a ZD7288-sensitive current obtained by subtraction.

The ZD-sensitive current was measured by subtracting traces obtained following ZD7288 from respective baseline values. The ZD-sensitive current was larger in delayed compared to immediate firing motoneurons measured both at -70 mV and -110 mV. These data are summarized in the text on page 12, Lines 388-391.

2) An original study (Leroy et al., 2014) previously showed in neonatal mice that the input conductance is smaller, the rheobase is lower and the voltage threshold for spiking is more hyperpolarized in immediate firing motoneurons (slow MNs) than in delayed firing ones (fast MNs) during the 2nd postnatal week. Furthermore, the study reported that delayed firing motoneurons have larger soma size and dendritic length than immediate firing motoneurons. The results of this study should be clearly stated in the discussion.

We have highlighted similarities between our findings and the findings of previous studies (including the work of Leroy et al., 2014) in the Results section (Lines 104-106) and in the second paragraph of the discussion on Lines 516-528. We have also included a supplementary, summary table, which directly compares key properties across studies. This new table (Supplementary File 3) has been referenced in the discussion.

3) In cats, the threshold of persistent inward currents has already been reported to be more hyperpolarized in small MNs and can be referred (Lee and Heckman, 1998).

This has been highlighted in the discussion on Page 19, Line 591-596.